# Observations of the seiche that shook the world

Thomas Monahan [1] ✉, Tianning Tang [1,2,3], Stephen Roberts[1,3] & Thomas A. A. Adcock[1,3]

On September 16th, 2023, an anomalous 10.88 mHz seismic signal was observed globally, persisting for 9 days. One month later an identical signal appeared, lasting for another week. Several studies have theorized that these signals were produced by seiches which formed after two landslide-generated mega-tsunamis in an East Greenland fjord. This theory is supported by seismic inversions, and analytical and numerical modeling, but no direct observations have been made. Here, we present primary observations of this phenomenon using data from the Surface Water Ocean Topography mission. By ruling out other oceanographic processes, we validate the seiche theory of previous authors and independently estimate its initial amplitude at 7.9 m using Bayesian machine learning and seismic data. This study demonstrates the value of satellite altimetry for studying fast oceanic processes and extreme events, while also highlighting the need for specialized methods to address the altimetric data's limitations, namely temporal sparsity. These data and approaches will help in understanding future unseen extremes driven by climate change.

Extreme events are evolving as a direct consequence of climate change, leading to the emergence of new, previously unobserved phenomena[1,2]. In remote regions like the Arctic, where in situ measurements are sparse, scientists depend on analytical and numerical models to explore these events. However, modeling in such regions presents significant challenges due to the uncertainties in the data required to calibrate and validate these models[3]. Consequently, large simplifications are often necessary, potentially resulting in substantial discrepancies between observed and modeled phenomena.

The mysterious 10.88 mHz very-long-period (VLP) seismic signal, which appeared following a tsunamigenic landslide in the Dickson fjord, Greenland, on September 16th, 2023, and the subsequent interdisciplinary scientific efforts to determine its origin, underscore these challenges. Two independent studies[4,5] have hypothesized that the signal was driven by a standing wave, or seiche, which formed in the aftermath of the tsunami. While it is well-documented that seiches can form in resonant enclosed and semi-enclosed basins[6], the loading-induced tilt they produce has only been observed locally (<30 km) and

for short durations (<1 h)[5,7]. Moreover, no prior evidence exists of persistent fluid sloshing (lasting several days) without an external driver. The 9-day attenuation of the globally detected VLP seismic signal is thus highly anomalous. Even more curious was the reappearance of the signal on October 11th, 2023, as noted by ref. 5, with approximately half the magnitude and duration of the initial event. This recurrence coincided with a second tsunamigenic landslide in the same gully in the Dickson fjord.

The rock-ice avalanches and subsequent tsunamis from both events have been well-documented through a combination of satellite and field observations, including evidence of tsunami run-up within the fjord and as far away as the research station at Ella ⌀, over 72 km away[4,5] (Section "Tsunami Information", and Supplementary Fig. 1). In contrast, evidence for the seiche has, by necessity, relied solely on a combination of analytical and numerical models, supplemented by seismic observations. While the seismic amplitudes are well reproduced, significant discrepancies remain between these studies in the estimated initial amplitude of the seiche (ranging from 2.6 m to

[1]Department of Engineering Science, University of Oxford, Oxford, UK. [2]Department of Mechanical and Aerospace Engineering, University of Manchester, Manchester, UK. [3]These authors contributed equally: Tianning Tang, Stephen Roberts, Thomas A. A. Adcock. ✉e-mail: thomas.monahan@eng.ox.ac.uk

7.4–8.8 m). Even more perplexing is the fact that the earth-shaking seiche was not observed by the Danish military whilst surveying the fjord on September 19th[5]. Hence, empirical observations are essential not only to confirm the existence of the seiche but also to validate the models and refine our understanding of the event dynamics.

To support the conclusions that the 10.88 mHz VLP signal was produced by a seiche in the Dickson fjord the authors of ref. 4,5 applied two independent approaches. Both studies identify the observed radiation pattern of seismic Rayleigh and Love waves to be consistent with an oscillating single force perpendicular to the Dickson fjord. Seismic Rayleigh waves exhibit elliptical, rolling motion with both vertical and radial displacement, propagating radially from the source. Love waves exhibit purely transverse motion perpendicular to their propagation direction. Additionally, both groups identify seismic source locations near the Dickson fjord. Here we present a much abbreviated summary of the additional methods and evidence.

By performing a seismic inversion on three sets of teleseismic arrays, the authors of ref. 4 isolate a predominantly horizontal and perpendicular force to the Dickson fjord. This finding serves as the basis of a simple analytical model of the sloshing physics by considering a simplified rectangular fjord geometry of $2 \times 20$ km. Using the single force inversion, they identify an initial horizontal force of ~160 GN, which leads to an estimated initial amplitude of 2.6 m. While a 2-d finite difference modeling approach was applied in the simplified rectangular geometry, this model only serves to illustrate that the fundamental-mode oscillation can form. Much effort is devoted to identifying the physical drivers of the observed damping, which, due to our approach not requiring these, we do not discuss further.

The large interdisciplinary team in ref. 5 use a combination of high-resolution numerical simulation and analytical modeling to corroborate their claims. Two numerical approaches are considered, however, their preferred approach employs a nonlinear hydrostatic model implemented in HySea[8] which treats the landslide as a granular flow. Using a fine grid spacing of 3 m, the simulation stabilizes into a slowly decaying seiche after ~5 min with an initial amplitude of 7.4 m. Notably, the first eigenmode has an oscillation frequency of 11.45 mHz (85 s), which differs from the observed VLP signal frequency of 10.88 mHz (92 s). This numerical simulation is then employed as a source time function to generate global seismic waveforms. Through direct comparison of the simulated envelope, the authors find good agreement between the synthetic and observed signal attenuation. An analytical model is also utilized, using a more realistic simplified geometry than in ref. 4. The authors identify the initial force to be 500 GN, significantly larger than the 160 GN estimate in ref. 4. This yields an estimated initial seiche amplitude of 8.8 meters.

While both studies provide compelling evidence that the source of the persistent 10.88 mHz signal was a seiche originating in the Dickson fjord, significantly different values are obtained for the initial amplitude of the seiche (2.6 m vs. 7.4–8.8 m). Both studies attribute these discrepancies to unmodelled effects. We note that both studies consider significantly different simplified fjord geometries, e.g.,[4] assume a fjord width of 2 km, and a length of 20 km, and[5] assume a width of 2.7–2.88 km (depending on the figure) and a length of 10 km. Naturally, these choices will lead to different analytical estimates of seiche amplitudes.

Here, we offer a completely different approach—using primary observations of this phenomenon to answer these questions. Using these data in conjunction with seismic observations, we estimate initial amplitudes of the September and October seiche events and compare these results with the prior studies. These observations and estimates provide conclusive evidence for the existence of this phenomena.

## Results
### Surface Water Ocean Topography (SWOT) mission
In contrast to in-situ devices, satellite altimetry provides near-global measurements, albeit with an inevitable trade-off in temporal sampling[9]. After more than 30 years, these data have revolutionized our understanding of many oceanic and climatic processes[10]. However, significant challenges arise in the study of extreme events due to a combination of the temporal sparsity, and the 1-d nature of the observations. Conventional Nadir altimeters sample data directly beneath the spacecraft, producing 1-d profiles along the sea-surface. This sampling severely limits the ability to draw conclusions regarding the spatial dynamics of extremes, and often leaves events unobserved altogether.

The wide-swath Surface Water Ocean Topography mission, launched on December 15th, 2022, has overcome many of these deficiencies[11]. Unlike conventional Nadir altimeters, the KaRIn instrument onboard SWOT provides ultra-high resolution 2-d measurements of ocean surfaces extending 50 km on either side of the spacecraft[12]. SWOT provides high accuracy measurements directly up to coastlines, and uniquely into fjords, with an effective pixelcloud resolution of 2.5 m along-track and a variable resolution ranging from 10 m to 70 m in the cross-track direction[11]. These measurements have low instrumental noise of less than 0.4 cm[13]. An overview of these sampling characteristics and an example of a single SWOT pass over the Dickson fjord is shown in Fig. 1C. The pixelcloud contains more than 300,000 measurements and provides complete coverage of the study region.

After transitioning to the Science orbit phase of the SWOT mission, SWOT made several observations of the Dickson fjord shortly after the occurrence of both tsunamigenic landslides. For the September 16th event, these passes occurred 0.5-days, 1.5-days, and 4.8-days after the VLP developed. For the October 11th event, only a single "usable" pass existed 0.5 days after the VLP began.

### Empirical observations
SWOT pixelcloud observations of the Dickson fjord for both the September and October events are shown in Fig. 2. For the October 12th observation (0.5 days post-tsunami), a large negative cross-channel slope (relative to the line $\overline{X_1 X_2}$) can be observed across the minor-axis of the fjord. Here we refer to the longitudinal-axis of the fjord as the major axis with the minor axis sitting perpendicular to it. While some noise artifacts exist around 27°W, the spatial distribution of resonant nodes is in good agreement with the high-resolution tsunami simulation in ref. 5.

Unfortunately, strong noise artifacts muddle large portions of the September 17th and 18th observations. However, on September 17th (0.5 days post-tsunami), a cross-channel slope can be observed anti-phase to both the October 11th and September 18th observations. Due to the noise artifacts on September 18th, estimates of the slope are not accurate. A description of the possible sources of these artifacts is given in Section "SWOT Data and Processing". The September 21st observations exhibit almost no noise artifacts and a very shallow negative cross-channel slope. The spatial distribution of nodes is consistent with those observed on October 12th and the tsunami simulation in ref. 5. We note that significant discrepancies exist between the simulated and observed sea-surface, particularly in the upper reaches of the fjord. Here, the SWOT observations suggest the presence of a persistent water build-up which appears to go unmodeled due to boundary choices.

### Seismic attribution
The SWOT data alone cannot estimate the total amplitude of the seiche as the observations could have occurred at any phase of the oscillation period. To overcome this, SWOT observations are referenced to seismic data from the II.ALE seismic station located at 82.5033°N

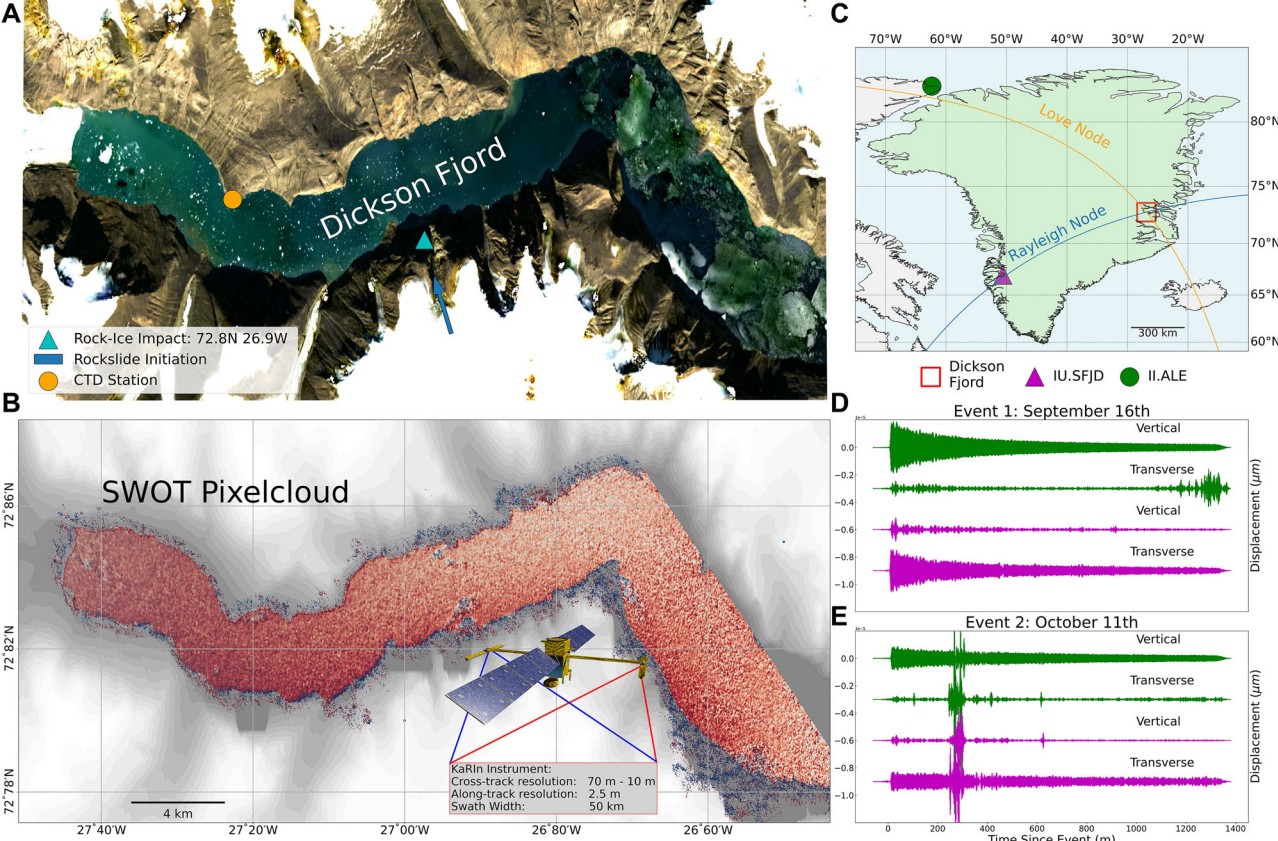

**Fig. 1 | Dickson fjord study region, Surface Water Ocean Topography mission (SWOT) measurements, and in-situ measurements. A** Sentinel-2 image of the Dickson fjord in summertime, with rock-slide, conductivity, temperature, and depth (CTD) gauge, and atmospheric station shown (Copernicus Sentinel data 2023). **B** Visualization of the study region (Made with Cartopy Natural Earth[39]), and nearby IU.SFJD and II.ALE seismic stations. Rayleigh and Love nodes are plotted in blue and orange. **C** SWOT pixelcloud measurements from a single pass over Dickson fjord, measurements colored by measured sea-surface height. SWOT satellite courtesy of NASA/JPL-Caltech. Seismic observations at II.ALE (green)[24] and IU.SFJD (magenta)[25] bandpass filtered between 10 and 13 mHz for the September 16th (**D**) and October 11th (**E**) events. Time units are minutes.

62.3500°W (1322.9 km away). The II.ALE station sits directly adjacent to the Love node and is thus characterized by almost exclusively Rayleigh waves as shown in Fig. 1B, D. The measured vertical displacement at II.ALE is filtered between 10 and 13 mHz, and shown for both events in Fig. 3A, B. SWOT observations are shown as vertical lines. We estimate a phase speed of 4.03 km s$^{-1}$ using a heterogeneous Earth model[14], and an approximate distance from the seismic source of 1409.5 km computed using the same heterogenous Earth model (see Section "Seismic Attribution" for details, and Supplementary Fig. 1 for relative location). Estimates of the uncertainty in this value, as well as validation using synthetic data are also provided in Section "Seismic Attribution". The relative magnitude and phase of the seiche can be directly determined through comparisons with the observed ground motion. Snapshots of the observed vertical VLP signal with SWOT observations highlighted are shown in Fig. 3C–F. The SWOT observed cross-channel slopes, computed between points $X_1$ and $X_2$ which gave rise to these signals are plotted directly below in Fig. 3G–J. The observed cross-channel slopes nicely correspond to the vertical displacement produced at station II.ALE. That is, negative cross-channel slopes (from $X_1$ to $X_2$) are associated with a negative vertical displacement and vice versa. The magnitudes also show good agreement. We note that this is exactly what is expected from the horizontal force produced by the seiche oscillation (Section "Simple Analytical Seiche")[5].

To validate these observations, the normalized VLP signal is used to estimate the initial amplitude of the seiche. Estimated slopes are computed using a Bayesian linear model (Section "Bayesian Regression"). Uncertainty estimates are obtained at both the parameter level,

alongside estimates of the noise content of the data. Due to the fact that each study assumes a different width and length of the fjord, we instead consider each estimate in terms of the corresponding cross-channel slope at maximum amplitude (MXCS). For[5] their initial amplitude estimate of 7.4–8.8 m translates to an MXCS of 2.56–3.13 m km$^{-1}$, and the initial amplitude estimate of 2.6 m by ref. 4 yields an MXCS of 1.3 m km$^{-1}$. Using the SWOT observations from September 17th we estimate the MXCS to be 1.83 ± 0.59 m km$^{-1}$. This value sits in between the estimates of both previous studies, however, the analytical estimate by ref. 4 sits at the very bottom of the confidence intervals, and the numerical estimate from ref. 5 just outside the top. Data from September 18th and 21st were not utilized due to large relative uncertainties and low Bayesian $R^2$ scores[15] (see Section "Bayesian Regression"). For the October event, we estimate the MXCS to be 1.37 ± 0.13 m km$^{-1}$. The fact that the October 12th observations occurred near a local minimum in the seiche's oscillation allows for tighter uncertainty estimation.

## Ruling out other suspects

While the SWOT data provides an unprecedented look at the instantaneous water levels in the Dickson fjord, it is only that; a snapshot. The observed cross-channel slopes conform to our expectation of a standing wave oscillating perpendicular to the major-axis of the fjord. However, there are other geophysical phenomena which can give rise to large cross-channel slopes in enclosed basins, namely tides[16] and wind-driven circulation (Ekman transport)[17]. Consideration is given to each of these possible causes.

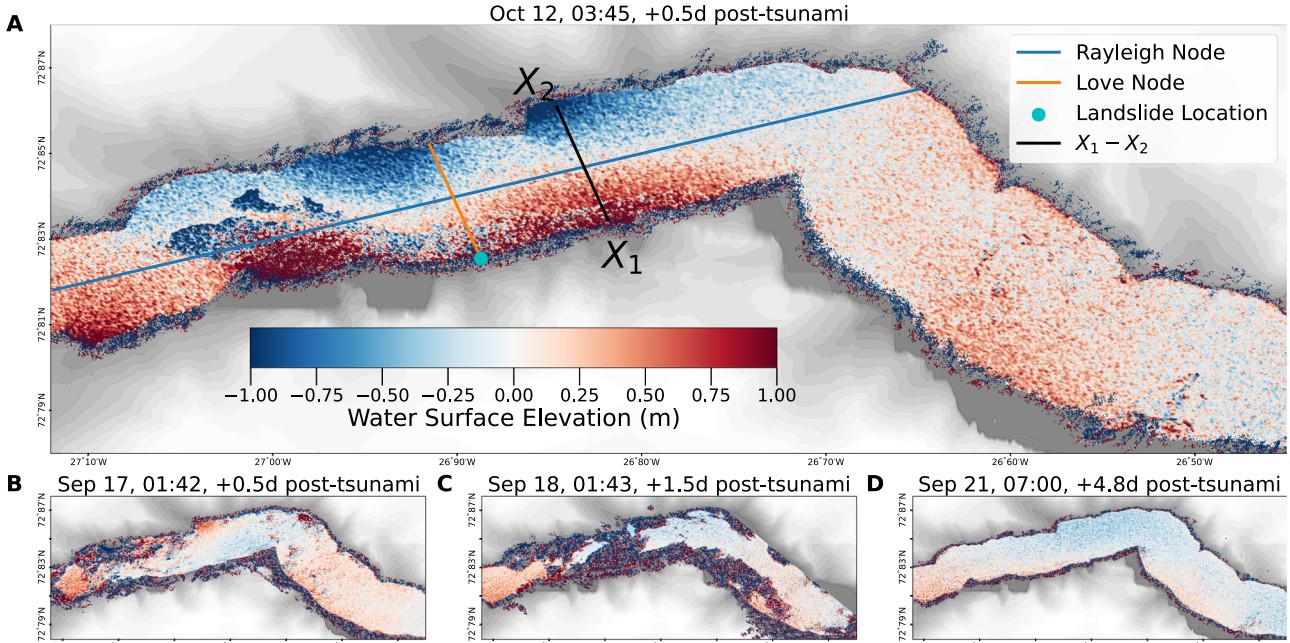

**Fig. 2 | Pixelcloud sea-surface elevation maps of the Dickson fjord in the days following the two tsunamis. A** Surface Water Ocean Topography mission (SWOT) observation of the fjord 0.5 days after the October 11th tsunami. Rayleigh and Love nodes are overlaid to show the theorized axis of propagation. **B, C, D** Consecutive SWOT observations of the fjord 0.5 days, 1.5 days, and 4.8 days after the September 16th event respectively.

**Tides.** While tides are obviously not the source of the ~92-s seismic signal, due to the sparsity of the SWOT measurements, tidally driven cross-channel slopes could lead to false conclusions about the presence of a seiche in the SWOT data. As there is only one tide gauge in the region, insufficient in situ information exists to rule this out. Additionally, small fjords are poorly resolved by state-of-the-art global tide models due to interpolation and can thus not be relied upon[18,19]. We instead directly analyze the SWOT pixel cloud data obtained over the year following the two events using a spatially coherent Bayesian harmonic analysis procedure[20]. This procedure has been shown to improve tidal estimation over conventional least-squares approaches for sparse reference series and in complex coastal regions. A complete description of our implementation is given in Section "Tidal Estimation".

Figure 4 shows empirical estimates of the amplitude and phase lag of the dominant lunar ocean tide, M2. Pixelcloud data is obtained and analyzed from October 20th, 2023 to November 1st, 2024. Due to the presence of winter sea ice and errors in the SWOT data, we identified only 23 usable passes. This limited data is insufficient to resolve additional tides, however, suffices for M2. As expected, both amplitude and phase exhibit small linear trends along the major-axis of the fjord and estimates are in good agreement with the CTD gauge (Fig. 4). Along the cross-channel direction, a slight rise in amplitude (5–10 cm) is seen in the center of the fjord but tapers off closer to the shores. This behavior may influence the curvature of the cross-channel profile, but will not induce the linear slope observed. The derived phase lags are uniformly distributed along the minor-axis of the fjord. Due to the fact that cross-channel variations in phase lag give rise to cross-channel slopes, we conclude that the observed slope in the SWOT data is not tidally driven.

As noted in ref. 4, an ~6 h modulation of the seismic signal can be observed. While this appears tidally correlated, the behavior is actually opposite of what is expected from a simple traveling wave. A proposed mechanism for this is given in Section "Tidal Modulation of the Seiche".

**Ekman transport.** Ekman transport occurs as a consequence of sustained wind-stress[21]. In the Northern hemisphere, the Coriolis effect causes water to propagate 90° clockwise to the incident wind direction. We evaluate the wind-speed and direction from the CTD atmospheric station shown in Fig. 1A. Results are shown in Fig. 5. Observations over the duration of the first VLP signal in September show that all SWOT measurements occur after sustained periods of southerly winds at ~5 knots. This wind-stress can induce a build-up of water in the westerly direction, but should not induce a cross-channel slope. The observations on October 12th follow yet another sustained period of Southerly wind, this time with a higher magnitude of ~10 knots. The SWOT observations occur as the wind is switching direction at low magnitudes (<5 knots). While sustained winds in the North-Western direction could give rise to cross-channel slopes, the low magnitude and short-duration of this change are unlikely to give rise to the large (2 m) cross-channel slope observed.

## Discussion

Our study provides direct observational evidence of the seiche in the Dickson fjord. Based on the seismic attribution, and systematic ruling out of other dynamic phenomena, we conclude that the observed variability in the SWOT data is consistent with that of a slowly decaying seiche. Thus, this study corroborates the numerical and analytical evidence given in ref. 4, 5, that the globally observed VLP signal which originated on September 16th, 2023, was due to a seiche which formed after a megatsunami. Additionally, we conclude that the smaller VLP signal observed on October 11th, was also a seiche originating from a smaller tsunami in the very same fjord. Notably, a short period seiche persisting for several days without an external driver has not previously been observed.

Using an empirical and completely independent approach, we estimate the maximum cross-channel slope (MXCS) of the September seiche to be 1.83 ± 0.59 m km⁻¹. This value falls between the analytical estimate of ref. 4 at 1.3 m km⁻¹ and the numerical/analytical estimates of ref. 5, which range from 2.56 to 3.13 m km⁻¹. Due to the relatively large uncertainty, our empirical estimate–when considered in isolation–provides limited insight into the true initial value. However, our analysis of the October VLP signal produced an estimate of the October seiche's MXCS at 1.37 ± 0.13 m km⁻¹. The smaller uncertainty in

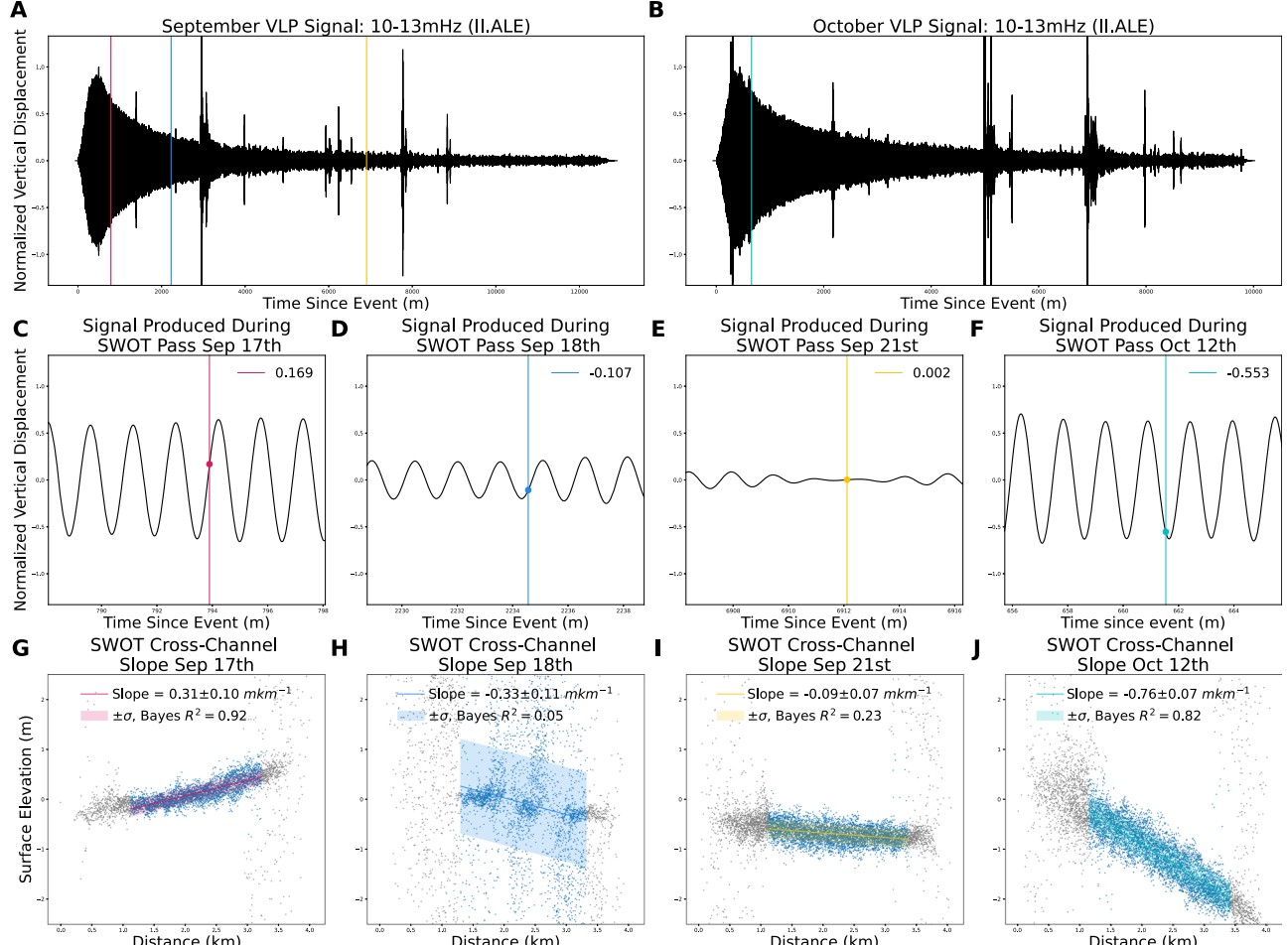

**Fig. 3 | Seismic observations of September and October very-long period (VLP) signals at II.ALE Seismic Station and Surface Water Ocean Topography mission (SWOT) cross-channel slopes. A, B** Normalized vertical ($\hat{Z}$) displacement VLP signals filtered between 10 and 13 mHz for the September and October events, respectively. SWOT observations are given by vertical lines. **C–F** Normalized vertical ($\hat{Z}$) displacement VLP signals with signals observed by SWOT shown as vertical lines. Observed magnitudes relative to the maximum amplitude are shown. **G–J** Corresponding SWOT cross-channel observations from $X_1$ to $X_2$. Slope estimates and associated Bayesian $R^2$ values from a Bayesian linear model are provided (Section "Bayesian Regression").

this case, owing to the October 12th observation occurring near a local minimum of the vertical displacement, gives greater confidence in this estimate.

Seismic data from the II.ALE and IU.SFJD stations show that the initial magnitude of the October VLP signal was approximately twice that of the September event (3 μm vs. 1.5 μm at II.ALE). Since the horizontal force is proportional to the average cross-channel slope (see Section "Simple Analytical Seiche"), we made a second estimate of the September MXCS to be 2.74 ± 0.26 m km⁻¹. This value closely aligns with the numerical/analytical estimates of 2.56–3.13 m km⁻¹ reported by ref. 5. Therefore, based on the relative magnitudes of the two forces at adjacent seismic stations, and the robust estimate of the October 12th event, we conclude that the numerical and analytical estimates by ref. 5 are in good agreement with the real data. Furthermore, we argue that the estimate provided in ref. 4 likely underestimates the true magnitude due to inaccuracies in the assumed fjord geometry, and the underestimation of the initial force at only 160 GN. If we consider a mean fjord width of 2.88 km as in ref. 5, our empirical estimate suggests the tsunami stabilizes into an initially 7.9 m tall seiche.

This study highlights the value of wide-swath satellite altimetry in characterizing extreme events and oceanic phenomena more generally. Due to their short-periods, seiches have long been difficult to study from conventional altimeters. The spatial resolution of SWOT provides opportunities in this area, as well as for studying other fast-

moving oceanic processes such as storm surge, and even large waves. As shown, these data also provide the opportunity to connect and understand the complex interactions between climate change and the different components of the geosphere. However, this work also emphasizes the importance of specialized, interdisciplinary methods to address the intrinsic limitations of these data, particularly the challenges posed by temporal sparsity. The SWOT data at its present level of processing, is not trivial to work with. Dedicated efforts are needed to improve reprocessing of these data in fjords. Additionally, open-source tools that bridge the gap between raw data and analysis pipelines are essential for enabling non-expert users to utilize these data.

While sufficient observational and bathymetric data existed to recreate the observed seiche dynamics numerically, for many remote regions this is not the case[3]. Indeed, while the effects of climate change are felt globally, the largest and fastest changes are often in these regions[22]. As such, while we echo the claims from previous authors about the need for more in-situ sea-level gauge sensors, we believe a follow-on to the SWOT mission and investment in future wide-swath and ideally non-sun-synchronous altimetry missions is also critical to monitoring these effects. Above all, designing orbits to achieve higher temporal sampling at lower latitudes should be prioritized as it presently limits the study of extremes. Furthermore, we stress that a key tool for identifying these events is the accurate computation of sea-

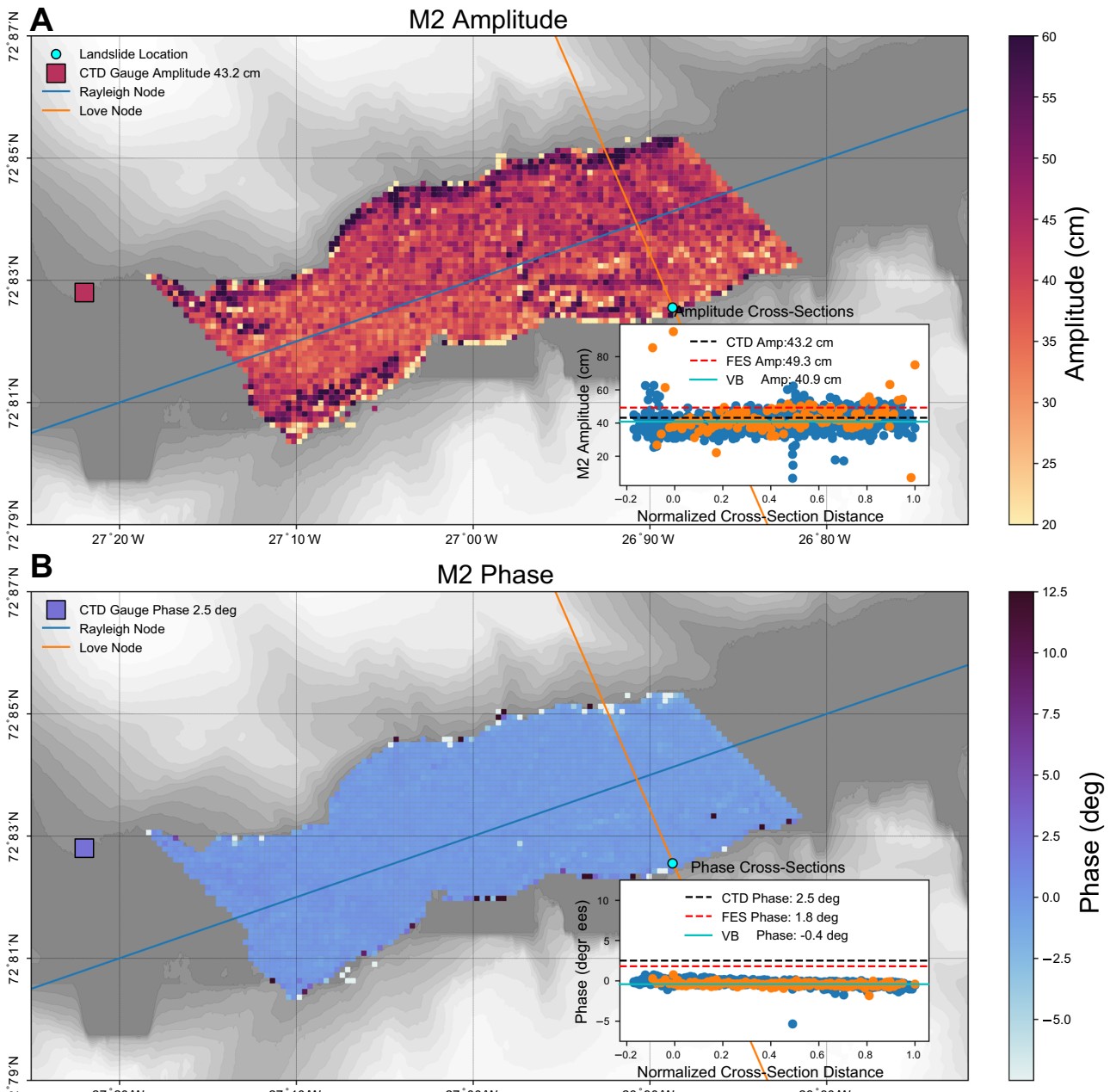

**Fig. 4 | Estimates of the dominant lunar tide, M2, from Surface Water Ocean Topography mission (SWOT) pixelcloud data using a spatially coherent variational Bayesian harmonic analysis (VB). A** Estimated amplitudes, and **B** corresponding phase lags. Estimates are only made for points which have at least 23 measurements. Inset plots show how the amplitude and phase vary along the Rayleigh (blue, left to right) and Love (orange, bottom to top) nodes respectively. The M2 amplitude and phase lag computed from the depth measurements at the conductivity, temperature, and depth (CTD) station are shown by the square. Horizontal lines showing the average VB, Finite Element Solution 2022 (FES), and CTD estimates are shown.

level anomalies (SLA) by applying different geophysical corrections. Due to the complex and narrow geometries of fjords, and a lack of historical altimetric measurements, tidal estimates in these regions are poor[18]. Additionally, we find that even when accurate SLA estimates can be obtained, the mean SLA of the fjord does not exhibit signs of seiching. This underscores the importance of accounting for small-scale spatial variances which SWOT and future wide-swath altimetry missions have the potential to address.

## Methods
### SWOT data and processing
Pixelcloud data from the SWOT mission are obtained through open Earth Access (last accessed on February 3rd, 2025). We utilize the version 2.0, HR pixelcloud data with shortname "SWOT_L2_HR_-PIXC_2.0" in the earthaccess API. The SWOT high-rate pixel cloud data is also available at: https://doi.org/10.5067/SWOT-PIXC-2.0. We note that all data utilized in the present manuscript is included in the replication materials also. At the time of access, the level 2.0 data constitutes the highest processing level available. The Dickson fjord landslides in September and October of 2023 occurred shortly after the transition of SWOT to the Science phase. The SWOT Science phase is characterized by a 20.86 day repeat orbit with 10-day sub-cycles. The orbit is at an inclination of 77.6°, and is thus non-sun-synchronous, which reduces tidal aliasing. A consequence of the 10 day sub-cycles is that repeat measurements occur in groups leading to relatively short measurement gaps at some portions of

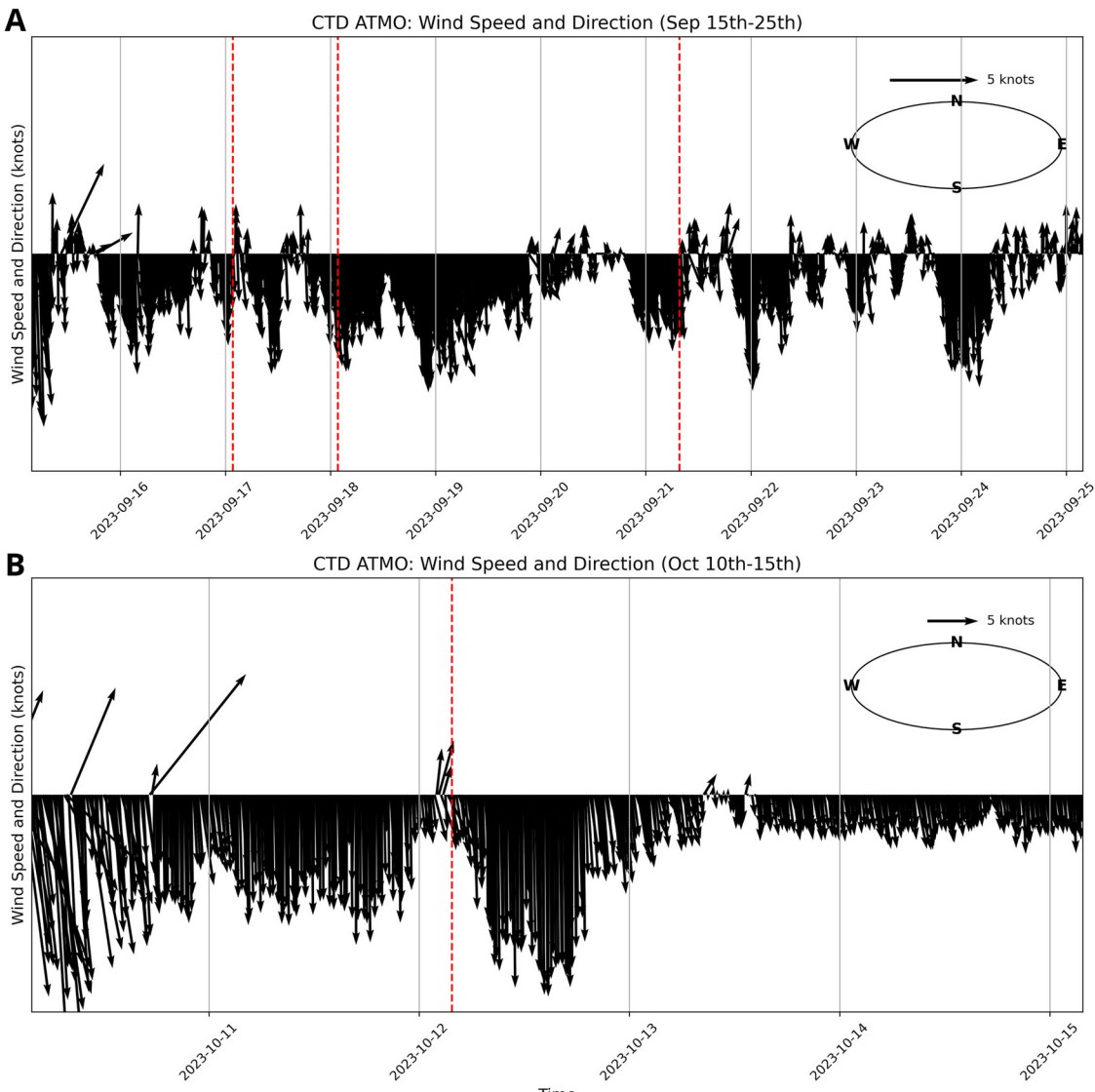

**Fig. 5 | Wind speed and direction from the Dickson fjord conductivity, temperature, and depth (CTD) station during each event. A** Wind speed and direction over the duration of the September 16th very-long period (VLP) signal **B** Wind speed and direction over the duration of the October 11th VLP signal. The red dashed vertical lines indicate when Surface Water Ocean Topography mission (SWOT) observations occur.

the cycle, and complete gaps in the other. Data for the September and October events were obtained by filtering all available passes between September 16–26, and October 10–18th, respectively. We select all passes which fall into the bounding box given in Table S1. For the September period, this yields 5 initial passes occuring 0.5 days, 2 days, 3 days, 4 days, and 5 days after the event, respectively. Similarly, for the October period, 2 initial passes are identified at 0.5 days and 6 days post-tsunami. We process the SWOT data in two stages, first through manual inspection and then using a standard preprocessing procedure.

**Manual inspection.** Prior to processing each pass we perform a manual inspection. Due to limitations in the L2 processing algorithms applied to SWOT data, entire passes can be contaminated to the point where the data becomes unusable. An example of this contamination can be seen in Supplementary Fig. 2. These errors are visually obvious and are manually flagged and removed through visual inspection. This yields a final set of observations for the September event 0.5 days, 2 days, and 5 days, and only a single observation for the October event taken 0.5 days post-tsunami.

**Data processing.** All standard geophysical corrections are applied including wet and dry troposphere delays and cross-over corrections. Unlike the Cal/Val SWOT orbit, the cross-over correction for the Science phase is much more accurate due to the significant reduction in time between crossovers. Geoid corrections are applied based on the EGM2008 geoid model. Due to interpolation errors with the default FES2022 tidal corrections, a specialized approach is needed to handle tides. A comprehensive discussion on how tides are dealt with is given in Sections "Tidal Estimation" and "Tidal Validation". After applying the geophysical corrections, we filter out all measurement values with sea-surface heights exceeding ±4 m. This threshold was empirically determined following manual inspection of the dataset employed for tidal estimation. Due to retracking issues, spurious observations exist over the fjord which exceed this 4 m threshold and are thus filtered out to avoid biasing estimates. To obtain perceptually uniform sea-surface height maps which can be compared between passes, we set the center of each colormap to be the midpoint of the cross-channel slope. Due to small inaccuracies with the tidal corrections described in Sections "Tidal Estimation" and "Tidal Validation", we found this procedure to be more robust.

**Cross-channel slope computation.** Due to the presence of noise artifacts in some of the SWOT measurements, we rely on estimates of the cross-channel slope between the points $X_1$ and $X_2$ for our analysis. We note that the slice between these points is perpendicular to both the long-axis of the fjord, and the Love node shown in Fig. 2. As a consequence of the nonuniform sampling of the pixelcloud data, it is necessary to define a tolerance which defines how far a point can be from the defined cross-section to be included. We experimented with several tolerance values but converged on a value of 333 m on either side of the defined cross-section. An example of this is shown in Supplementary Fig. 3, with ~8000 measurements shown. This choice provides a representative 666 m swath (≈5% of the fjord length) with which to estimate the variability/uncertainty of the cross-channel slope. A description of the Bayesian linear model employed for this procedure is given in Section "Bayesian Regression".

**SWOT data for tidal estimation.** Pixelcloud data is also used to estimate the M2 tide throughout the Dickson fjord. Due to the severe aliasing produced by the irregular temporal sampling, we make use of all available data. The presence of winter sea ice creates further difficulties as the inclusion of these data will severely degrade the accuracy of tidal estimation. As before, we locate all passes which intersect with the study region (Table S1) between October 20th 2023, and November 1st, 2024. This yields an initial dataset of 241 passes. After manual filtering and removal of passes containing sea-ice, this leaves only 23 high-quality passes which are employed for tidal estimation. We apply the same geophysical corrections as before. In order to obtain time-series which can be fed to our spatially coherent Bayesian harmonic analysis we bin the nonuniformly sampled pixel cloud data into a fixed grid with a 100 m × 100 m resolution. Experiments were conducted at variable resolutions and this was found to yield a good balance between high resolution, and reduced sensitivity to noise. Data points in each bin are averaged. The total number of data per bin is variable, thus, we restrict our analysis to bins which contain at least 23 measurements. This threshold is empirically defined based on the fact that the usage of fewer than 23 measurements in testing produced spurious tidal estimates in some regions.

## Seismic data

Seismic data are accessed using the Federation of Digital Seismic Networks web service client for Obspy[23]. As both[4] and[5] provide a comprehensive analysis of the long-period seismic signal, we choose to focus on two representative stations: II.ALE[24], and IU.SFJD[25]. In order to isolate the energy associated with the VLP, data are shown after being bandpassed between 10 and 12 mHz. All additional filtering parameters needed to replicate our results are provided in the "seismic_attribution.ipynb" notebook.

## Seismic attribution

Seismic attribution is performed using data from the II.ALE seismic station. The station is situated 1322.9 km away from the landslide source at 82.5033°N 62.3500°W. The station sits directly adjacent to the Love node (160°) which runs perpendicular to the major-axis of the fjord (Fig. 1B). As a consequence, this station receives almost exclusively Rayleigh waves which is reflected by the dominant vertical component of the VLP signal shown in Fig. 1D. Referencing the SWOT observations to these waveforms requires the accurate identification of the phase velocity and source location of the VLP signal.

**Phase velocity.** To compute the Rayleigh wave phase velocity, we utilize a heterogeneous Earth model as in ref. 5. To obtain the phase velocity of the 10.88 mHz signal we interpolate from the 10 mHz and 15 mHz LITHO1.0 velocity models[14]. Integrating over the path between the Dickson fjord and the II.ALE station we obtain a mean phase velocity of 4.0393 km s⁻¹.

**Source location.** Due to uncertainties in the computed phase velocity, the source location of the VLP signal is not necessarily given by the center of the Dickson fjord. Indeed, both[4] and ref. 5 identify source locations which are nearby, but away from the fjord itself. Using the same heterogeneous Earth model and a Fast Marching method for beamforming, a source location 92.4 km from the landslide location at 72.2°N 25.1°W is identified[5]. This source location is ~1409.5 km away from the II.ALE seismic station. Combined with the computed phase velocity, this yields a travel time of 348.97 s.

**Validation with synthetic observations.** To test our hypothesis that the phase velocity and source location obtained using the heterogeneous Earth model are appropriate for seismic attribution we compare our approach to an independent forward modeling exercise carried out in ref. 5. Using their 3 m HySEA simulation of the seiche as a source time function, synthetic 3-component Green's functions are computed using the Syngine web service and convolved to approximate the displacement signals at the II.ALE station. In order to align the two signals, the authors identify an ~350 s shift empirically. This value is in agreement with the 348.97 s travel-time computed using the heterogeneous Earth model and beam forming. This agreement between two completely independent methods confirms the validity of the 348.97 s Rayleigh wave travel time to II.ALE for seismic attribution.

**Seismic uncertainties.** While our estimated Rayleigh wave travel time is in good agreement with the empirical estimate by ref. 5, this represents an important source of uncertainty in the estimation of the seiche amplitude. No comprehensive approach exists for the quantification of Rayleigh wave phase velocity or beamforming location uncertainties[26]. As such, we provide a lower-bound on our uncertainty estimate using the discrepancy between our estimate of 348.97 s and the empirical estimate of 350 s by ref. 5. If we assume a 92-s period of oscillation, this yields an approximate error of 2.12 %. Further uncertainties exist due to the fact that near the point source, the phase is governed by a Bessel function rather than a complex exponential as utilized. Whilst the uncertainty this introduces is difficult to estimate, it is important that this is acknowledged[27]. Additionally, testing of homogeneous Earth models yielded different estimates of both source location and velocities, yielding variable results. We found the agreement between these models and the empirically estimated wave travel time to generally be worse and thus did not consider them further.

## Bayesian regression

Both cross-channel slope and tidal estimation are performed using a Bayesian linear model. Our selection of a Bayesian approach reflects our objective to accurately quantify the uncertainty in the estimated parameters and the SWOT data themselves. Unlike conventional least-squares estimation and other frequentist variants, which provide point estimates of the parameters, a Bayesian approach considers (and computes with) the probability distributions of the parameters. By representing our parameters of interest as probability distributions, the uncertainty associated with parameters in our model is explicitly represented, thus informing a calibrated measure of uncertainty over the target variables. As will be shown, the selection of appropriate priors can yield further advantages such as natural parameter shrinkage and increased robustness to noise. Shrinkage refers to reducing the magnitude of parameters which do not contribute to the solution. This favors simpler models unless the data justify additional complexity.

Consider the linear model given by $y_i = w^\top x_i + \epsilon_i$. Here, $y_i$ is the $i^{th}$ SWOT sea-surface height measurement, $w$ is a vector of the estimated weights (corresponding to the entries of the design matrix $X$), $x_i$ is the $i^{th}$ row of an $M \times N$ design matrix $X$, and $\epsilon_i$ is the residual. Here, $M$ is the total number of measurements, and $N$ is the number of input functions

(described below). Linear models of this form are employed for both cross-channel slope and tidal estimation. As will be described, in our Bayesian formulation, the inferred elements of the weight vector $w$ govern the shape of the various distributions used in the model. The parametric form for these distributions, and their justification, is given at length below alongside the variational inference procedure. The tidal estimation procedure is provided in Section "Tidal Estimation".

**Cross-channel slope estimation.** To estimate the cross-channel slope we perform a standard Bayesian linear regression on the instantaneous sea-surface height cross-sections from each pass. The $i^{th}$ row of the design matrix $X$ is simply given by $x_i = [1, x_i]$, where the entry 1 corresponds to the bias (intercept), and $x_i$ the distance along the line $\overline{X_1 X_2}$. Here, the observation $y_i$ is the corresponding SWOT measurement with distance $x_i$ along $\overline{X_1 X_2}$.

**Bayesian analysis.** Here we provide a brief overview of the Bayesian model utilized and the variational inference method. A complete derivation of variational inference can be found in ref. 28 and a more detailed exposition of the Bayesian linear model, used here, can be found in the appendix of ref. 29. The basis of our analysis is Bayes' theorem, which provides a framework for updating our prior beliefs $p(\theta)$ about the distribution of all parameters (and hyperparameters), based on a set of observations $Y = [y_0, y_1, \ldots, y_N]^T$ and the design matrix $X$. A hyper-parameter is itself a parameter, that governs the probability distribution of another parameter. For example, the mean and variance of a Gaussian are the hyperparameters which define the distribution over another variable. By extension, a hyper-hyperparameter describes a variable that controls the distribution over a hyper-parameter. This yields a posterior distribution which describes our final beliefs over $\theta$ and is given by

$$p(\theta|Y) = \frac{p(\theta)p(Y|X,\theta)}{p(Y)}, \tag{1}$$

where $\theta$ is the set of parameters and hyperparameters of the model, $p(\theta)$ is the prior, $p(Y|X, \theta)$ the (data) likelihood, and $p(Y)$ the marginal likelihood which acts as a normalizing term in the inference (as it does not depend upon $\theta$). Here we provide a brief overview of the different components of Bayesian analysis.

**Likelihood.** The likelihood $p(Y|X, \theta)$ describes the likelihood that the observed data occurred, given our model. Here, the model is defined by both the design matrix, and our prior assumptions about the parameters $\theta$. We take the likelihood term to be of Gaussian form, equivalent to a least-squares error assumption. In our analysis, we weight the squared residual between observed $y_i$ and model prediction, by a scalar hyper-parameter $\beta$, which represents the precision, or inverse (co)variance of the noise. Discussion and validation of this modeling decision in the context of SWOT based tidal analysis is provided in ref. 20. This has the effect of weighting how tightly our model should fit the data based on how noisy the data is. This assumption yields a Gaussian likelihood term of the form:

$$p(Y|X,\theta) = p(Y|X,w,\beta) = \left(\frac{\beta}{2\pi}\right)^{N/2} \exp\left\{-\frac{\beta}{2} E_Y(w)\right\} \tag{2}$$

where we see the likelihood only depends upon $w$ and $\beta$. The error functional $E_y$ is given by

$$E_Y = \sum_{i=0}^{N} (y_i - w^\top x_i)^2. \tag{3}$$

**Priors.** Central to Bayesian inference is the usage of priors. Priors are distributions over the parameters included in a model, and reflect our

initial expectation of the functional forms and values the parameters should have. Here we describe the choice of these priors for both parameters and hyperparameters, and describe how they impact the resultant model. Under the mean-field approximation, we assume the prior over all parameters $\theta$ can be factorized as

$$p(\theta) = p(w|\alpha)p(\alpha)p(\beta), \tag{4}$$

where $\alpha = \{\alpha_j\}$ is a set of hyperparameters which governs the scale of the multi-variate Gaussian over the weights $w$. We now treat each term individually.

The model weights $p(w|\alpha)$ come from a zero-mean Gaussian prior, with precisions (inverse variance) $\alpha$. This choice serves two purposes. First, a Gaussian is the least informative distribution for a quantity which can be either positive or negative, and is thus unbiased in this way[29]. This is important as both cross-channel slopes and quadrature tidal amplitudes can be positive or negative. Second, weights will only be significantly non-zero if the data requires it. Conventionally, this is referred to as an Automatic Relevance Determination prior, as it induces shrinkage over the model weights, which do not significantly aid the model in fitting the data. Using this, for an individual weight $w_j$, the prior has the form

$$p(w_j|\alpha_j) = \left(\frac{\alpha_j}{2\pi}\right)^{1/2} \exp\left\{-\frac{\alpha_j}{2} w_j^2\right\}. \tag{5}$$

The set of weight precisions $\alpha$, which govern the scale of the weights $w$, are drawn from a Gamma distribution, which models the distribution over non-negative precisions. In addition to non-negativity, this choice is made for several reasons. First, a Gamma hyperprior is conjugate with the Gaussian prior over the weights which will be useful when performing inference. A conjugate prior for a given likelihood function is a prior that results in a posterior distribution that is of the same family as the prior. A Gamma prior also implicitly encourages smaller weight magnitudes, leading to natural shrinkage and promoting sparsity in the inferred weights. Uninformative hyper-hyperparameters $a_0 = 10^{-2}$ and $b_0 = 10^{-4}$ which set the shape and scale of the Gamma are selected to yield a vague prior over each $\alpha_j$ defined as

$$p(\alpha_j) = \Gamma(\alpha_j; a_0, b, 0). \tag{6}$$

A vague or uninformative prior simply means the assumed distribution is broad. This imposes minimal assumptions regarding the parameter values, whilst still providing natural parameter shrinkage[30]. This is favorable to a uniform prior which implicitly restricts parameter values and can lead to improper posteriors.

The scalar noise precision $\beta$ (inverse variance) of the residual $\epsilon$ is also modeled as a hyperparameter within the model. Given the least-squares assumption of a Gaussian residual, we once again adopt a Gamma prior over $\beta$. The values of the shape and scale parameters are identical to those used in Equation (6), but are defined by the hyper-hyperparameters $c_0 = 10^{-2}$ and $d_0 = 10^{-4}$ such that

$$p(\beta) = \Gamma(\beta; c_0, d, 0). \tag{7}$$

**Initialization.** Models are initialized using a maximum-likelihood (ML) solution such that

$$w_{ML} = X^\top Y (X^\top X)^{-1}. \tag{8}$$

The ML solution is then used to initialize the residual precision hyperparameter, $\beta$, such that:

$$\beta^{-1} = \frac{1}{N} \sum_{i=1}^{N} (y_i - w_{ML}^{\mathsf{T}} x_i)^2 \qquad (9)$$

The ML solution provides an initial estimate that follows from the likelihood, ensuring that we start in an informative region of parameter space. This is useful for initializing the noise precision, $\beta$, and helps the variational updates from settling into poor local minima. While absolute guarantees depend on problem-specific properties, ML initialization is a well-established heuristic that improves the stability and efficiency of variational inference.

## Variational inference

Fully Bayesian solutions are obtained by marginalizing over the posterior distributions of the parameters. The difficulty arises when computing the posterior distribution, which analytically is almost always intractable. Hence, sample based approaches such as Markov-Chain Monte Carlo are often employed. While these methods are good at approximating the true posterior, they scale poorly with the number of parameters included. Further, convergence is not easily assessed. Here, we adopt an approximate inference approach, called variational Bayes, referred to herein as VB. VB is a computationally tractable alternative to MCMC, bringing both scalability and a principled approach to assess whether convergence has been achieved. These attributes are particularly important for the tidal estimation procedure, as with over 300,000 individual locations, assessing convergence manually is intractable. The objective of our analysis is to infer the distributions over the individual elements of $\theta$. The basic idea of VB is to adopt analytical approximations for each distribution, which can be optimized in an iterative and computationally tractable way. We first introduce an approximate posterior $q(\theta|Y)$. The functional form of this posterior is chosen to be conjugate with the prior over $\theta$ such that $q(\theta|Y)$ factorizes as

$$q(\theta|Y) = q(w|Y)q(\alpha|Y)q(\beta|Y). \qquad (10)$$

This choice follows from the mean-field approximation, which assumes our latent parameters to be mutually independent. This selection is deliberate, as it allows for analytical marginalization over the parameter posteriors, making inference computationally efficient. However, it also neglects correlations between parameters, potentially underestimating uncertainty in the presence of strong multicollinearity. Despite this, the approach has been extensively validated for both trend extraction and tidal estimation and is thus sufficient for these tasks[20,29].

Our objective in VB is to minimize the difference between the approximate posterior $q(\theta|Y)$, and the true posterior $p(\theta|Y)$. This difference can be assessed by considering our observable, the data evidence $p(Y)$. Using our approximate posterior, we can rewrite the log evidence $p(Y)$ as the sum of two separable terms such that

$$\log p(Y) = F(p(\theta|Y), q(\theta|Y)) + \mathrm{KL}(p(\theta|Y), q(\theta|Y)). \qquad (11)$$

This is the fundamental equation of VB and is composed of two terms. The first term is the negative variational free energy, referred to as the evidence lower bound. This provides a strict lower bound on the model evidence. The second term is the Kullback-Liebler divergence between the approximate and true posteriors over $\theta$. This term provides natural model shrinkage as it increases with the number of free parameters $\theta$. It can be seen that maximizing $F(p, q)$ will result in the approximate posterior being as close as possible to the true posterior. Due to the fact that $q(\theta|Y)$ can be factored as Eq. (10), $F(p, q)$ can be maximized by iteratively optimizing each of $q(\theta|Y)$, $q(\alpha|Y)$, $q(\beta|Y)$

separately. Update equations for this procedure can be found in ref. 31. An implementation of this approach can be found in the replication notebooks "Fjord_Tides.ipynb" and "seismic_attribution.ipynb".

**Bayesian R-squared.** To evaluate the quality of the Bayesian regression we utilize the Bayesian $R^2$ proposed in ref. 15. This is necessary as the variance of the predicted values, can be greater than the variance of the data, thus rendering the conventional $R^2$ definition nonsensical. The modified Bayes $R^2$ is simply given by

$$\text{Bayes } R^2 = \frac{\mathrm{Var(predicted)}}{\mathrm{Var(predicted) + Var(residual)}}, \qquad (12)$$

where Var(residual) is the expected variance of the errors as given by the model.

## Tidal estimation

Due to the extreme sparsity of available SWOT data (less than 25 measurements over a full year), extreme care is needed when performing tidal harmonic analysis[20]. Harmonic analysis assumes tides can be described by the superposition of waves at discrete tidal frequencies. These frequencies exist at harmonics of the motions between the Earth, Moon, and Sun and are described as constituents. In total, there are hundreds of possible constituents, however, for practical purposes we need only concern ourselves with a few. Given $n$ constituents, the $k$th constituent with frequency $\omega_k$ has a corresponding tidal wave of

$$C_k \cos w_k t - \phi_k = A_k \sin \omega_k t + B_k \cos \omega_k t. \qquad (13)$$

Comparisons of tidal constituents are done in terms of the amplitude $C_k = \sqrt{A_k^2 + B_k^2}$ and phase $\phi_k = \arctan A_k / B_k$. Modern tidal analysis is carried out in the time-domain using least-squares estimation, and can thus be applied to irregularly sampled time-series. To accomplish this we define the tidal estimation problem as a general linear model, such that the observed sea-level, $y_i$, at any time is given by $y_i = w^{\mathsf{T}} x_i + \epsilon_i$. Here, $x_i$ is the $i$th row of an $M \times N$ matrix of basis functions where $M$ is the number of measurements and $N = 2n + 1$ with $n$ equal to the number of constituents, $w$ is a set of inferred weights, and $\epsilon_i$ is the non-tidal residual. From this, we define a design matrix $X$ given by

$$X = [1, \sin \omega_0 t_i, \cos \omega_0 t_i, \dots, \sin \omega_k t_i, \cos \omega_k t_i]^{\mathsf{T}} \qquad (14)$$

where 1 corresponds to the bias, and the remaining values the quadrature amplitudes. The extreme sparsity of the Dickson fjord reference series is such that only the dominant lunar tide, M2, can be reliably estimated as shown in ref. 18. Even though we are only interested in M2, it is necessary to include additional constituents in the analysis to "soak" up the contributions of additional constituents[18,20]. After considerable experimentation, and validation against the in situ gauge measurements, it was found that the best M2 estimation was obtained by using the semi-diurnal M2, N2, S2, and diurnal K1, and O1 constituents in the analysis. Supplementary Table 2 provides the associated periods and origins of each constituent. Section "Tidal Validation", discusses validation with the in-situ CTD observations and the FES2022 tidal model.

Conventional harmonic analysis only considers measurements from a single spatial location. Due to the spatial coherence of the oceanic response to tidal forcing[32], this procedure leaves considerable information out. SWOT data provides a complete picture of the instantaneous sea-surface height throughout the Dickson fjord which can be exploited using an appropriate method. Here, we adopt the spatially coherent harmonic analysis procedure in ref. 20. Readers are referred to ref. 20 for a complete description of the procedure.

However, the basic idea is to simultaneously estimate the quadrature amplitudes across a set of adjacent points by assuming that the amplitude at any point $P_{j,k}$ is given by the amplitude $w_{0,0}$ of the central point $P_{0,0}$ with a small offset denoted $w_{j,k}$. The linear model can be expanded to

$$Y_{j,k} = X_{0,0} w_{0,0} + \rho X_{j,k} w_{j,k} \qquad (15)$$

where $Y_{j,k}$ is the observation at point $P_{j,k}$, $X_{0,0}$ and $X_{j,k}$ are the design matrices for points $P_{0,0}$ and $P_{j,k}$ respectively, and $\rho$ represents the probability that the observations $Y_{j,k}$ are correlated with $y_{0,0}$. The scalar $\rho$ is obtained by taking the Fischer transform of the Pearson's correlation coefficient $r$ between $Y_{0,0}$ and $Y_{j,k}$, such that $z' = .5[\ln(1+r) - \ln(1-r)]$, and $\rho = 1 - \Phi(z')$ with $\Phi = \mathcal{N}(0,1)$. By including the probability that the observations are correlated, $\rho$, we impose an assumption that points with more similar time-series will have similar tides. Furthermore, in the limit where $Y_{j,k} = Y_{0,0}$, the quadrature amplitude $Y_{j,k} = X_{0,0} w_{0,0}$. We can further restrict the functional form of the relationship between adjacent locations to boost the ratio of data to parameters. As validated in ref. 20, we approximate the relationship between quadrature amplitudes in adjacent locations to be linear, such that

$$Y_{j,k} = \rho \cdot (d_x \cdot X_{j,0} w_{j,0} + d_y \cdot X_{0,k} w_{0,k}) + X_{0,0} w_{0,0} \qquad (16)$$

where $d_x$ and $d_y$ are the normalized distances between points along the horizontal and vertical direction, respectively, taken to be (−1, 0, 1) for convenience when using gridded SWOT data. This approach effectively doubles the ratio of data to parameters and was found to improve uncertainty estimation[20]. By virtue of using gridded data with a resolution of 100 m, the assumption of linearity between neighboring quadrature amplitudes is more than sufficient.

While the spatially coherent harmonic analysis can be used in tandem with any estimator, here we make use of the variational Bayesian (VB) estimator described above in Section "Bayesian Regression". This choice is based on the following reasons. First,[20] find the VB approach to be less sensitive to both stationary (Gaussian) and non-stationary noise artifacts. Second, VB provides natural parameter shrinkage, which is helpful for reducing so-called crosstalk between constituents left out of the analysis. Lastly, the implicit uncertainty information is helpful in assessing the quality of the tidal estimates. The analysis shown in Fig. 4 only includes locations which have at least 23 observations. Supplementary Fig. 4 shows the distribution of SWOT measurements, which can be used for tidal estimation through the fjord. A complete implementation of this approach, and code to replicate all tidal estimation is given in "Fjord_Tides.ipynb".

## Tidal validation

Tidal validation is carried out for both the SWOT derived tidal estimates, and the standard tidal estimates from FES2022[33] using the in-situ CTD gauge as a ground truth[34]. For each constituent $k$, models (VB or FES) are assessed using the root-mean-squared error, defined as

$$\mathrm{RMS}_k$$
$$= \sqrt{\overline{[(A_{k,\mathrm{mod}} \sin(\omega_k t) + B_{k,\mathrm{mod}} \cos(\omega_k t)) - (A_{k,\mathrm{CTD}} \sin(\omega_k t) + B_{k,\mathrm{CTD}} \cos(\omega_k t))]^2}},$$
$$(17)$$

where the overbar indicates the average over a complete cycle (e.g., $0 \to 2\pi$). This metric reflects the combined error in predicted sea-level height introduced by both amplitude and phase errors. We also introduce a secondary metric, the relative RMS error, which weights the RMS error relative to the magnitude of the CTD gauge constituent

$\mathrm{RRMS}_k = \mathrm{RMS}_k / A_{k,\mathrm{CTD}}$. Comparisons between RMS and relative RMS error are provided in Supplementary Fig. 6. For the SWOT estimates, the median amplitude and phase for the entire fjord are used. As noted above, we only include the constituents M2, N2, S2, K1, and O1 in the analysis due to the limited SWOT data. For these constituents, it can be seen that M2 yields significant improvements over FES2022 of 52%. The SWOT derived N2, S2, K1, and O1 in contrast are relatively inaccurate, due to the smaller amplitudes and longer alias periods. Similar results have been found for early SWOT data by ref. 18. In their work additional constituents were also included to yield superior M2 accuracy. When correcting the SWOT data, we utilize just the M2 from SWOT, alongside S2, N2, O1, and K1 from FES2022. As can be seen in the relative RMS panel, many of the additional FES2022 constituents exceed 20% relative RMS error within the Dickson fjord. In order to avoid introducing potential biases from incorrect tidal corrections, we do not include constituents above this 20% threshold. To combat any residual tidal variability we instead define the center of the colormap to be the midpoint of the cross-channel slope.

It is important to note that there are large seasonal changes in tides within arctic regions[35]. Due to the limited SWOT data available, and the relatively short CTD gauge deployment (≈2 months), the extent of this variability is difficult to estimate. However, since the CTD gauge was deployed during both events, validation with this gauge is appropriate as we wish to have accurate tidal estimates during this period. As more SWOT data becomes available, the accuracy of tidal estimates will improve. It is unlikely that these improved estimates will change the conclusions made by this study, as we are primarily interested in the cross-channel slope, and find no evidence of this being tidally driven based on inspection of the M2 tide. However, improved tidal estimation could prove useful for understanding how the tide itself modulates the period of the seiche, and thus impacts its dynamics and dissipation.

## Simple analytical seiche

In order to estimate the total seiche amplitude, and to relate the September and October events using the observed seismic observations, we consider a theoretical seiche acting on a simplified fjord geometry. We adopt the notation and fjord geometry employed in ref. 5 for consistency. Here, we assume the seiche to act as an oscillating horizontal force directed N160°E (perpendicular to the Dickson fjord). Due to discrepancies between the defined geometries given in ref. 4, 5 we avoid prescribing values for the precise fjord dimensions except where necessary. The sloshing of the seiche produces a shift of the center of mass of the body of water $x$, and can be written as

$$x(t) = \Delta x \sin \omega t. \qquad (18)$$

Here $\Delta x$ is the amplitude of the horizontal oscillation and $\omega \approx 2\pi/92$ Hz is the frequency of oscillation. As described in ref. 5, the amplitude of the Rayleigh waves produced is proportional to the magnitude of the horizontal force. Hence, we write $\Delta x$ in terms of the total force $F$. Taking the second derivative of the position of the center of mass we find

$$F = \Delta x \omega^2 \sin \omega t. \qquad (19)$$

It can be seen that the maximum force occurs at the maximum displacement $\Delta x$ of the center of mass. Using SWOT data we can only observe the cross channel slope. As such, it is useful to relate the force back to the surface displacement $\Delta z$ such that

$$\Delta x = \frac{L}{3} \frac{\Delta z}{h + h_s} \qquad (20)$$

Equation (19) can be rewritten in terms of the vertical displacement $\Delta z$ as

$$F = \frac{Lm\omega}{3} \frac{\Delta z}{h + h_s} \qquad (21)$$

We recognize that the surface displacement $\Delta z = SL$ where $S$ is the cross-channel slope. Thus, the force $F$ can be written in terms of the cross-channel slope with

$$F = \frac{L^2 m\omega}{3} \frac{S}{h + h_s}. \qquad (22)$$

Thus, we have shown that the force is directly proportional to the cross-channel slope. This allows for the direct comparison between events.

### In situ measurements

In-situ measurements are provided by the CTD station located in the inner portion of the Dickson fjord (as shown in Fig. 1A and can be accessed at ref. 34). The station provides both standard meteorological and oceanic variables. The CTD gauge is mounted on the fjord wall, inside a HDPE tube to protect it from ice in winter. Here, we only make use of the wind speed and direction, density, and water depth measurements. Data are sampled at 15-min intervals, which creates severe aliasing issues for observing the 92 s VLP signal. Due to the location of the device in the inner fjord, the seiche signal magnitude decreases beneath pre-event noise levels after only a few hours and is thus unobservable in the data (See Supplementary Fig. 5).

### Dickson fjord

Here we present a brief overview of the Dickson fjord. A more comprehensive description of the physiography and climate of the fjord system can be found in ref. 5. The Dickson fjord sits at the terminus of the Hissinger Glacier in the northernmost area of the Kong Oscar fjord system situated in East Greenland (See Supplementary Fig. 1). The fjord itself sits deep in the Arctic Circle, and is thus characterized by sea-ice over much of the year. Sea ice dissipates in July and then forms again in October. The fjord fills a U-shaped valley basin, with multiple smaller glaciers situated on each side. The fjord itself is 38 km long and between 2.5 and 3.2 km wide. The depth ranges from 150–200 m to 700 m from West to East with an approximate depth of 540 m in the center of the fjord across from the landslide location. Bathymetry estimates are taken from a 2018 survey by the National Danish Hydrographic Office at a resolution of 15 m. We note that no data exists between 150 and 300 m of the coast due to the limitations of the survey vessel. This missing data creates large uncertainties in these regions, which can significantly influence numerical simulations.

### Tsunami information

Both tsunamis originated from landslides occurring in the same gully situated beneath an unnamed glacier[5]. These landslides were caused by debuttressing of the glacier following glacial thinning over the past decade. Direct observation of landslide scarring and dirtying of the glacier using satellite imagery by both[4,5] confirms this theory. Additionally,[5] evaluate the landslide dynamics of the September 16th landslide via seismic inversion. While the October 11th event did not produce a new landslide scar, a Sentinel-2 image showed significantly more erosion than was present after the September 16th event.

Empirical evidence of the two tsunamis is given by a combination of nearby in-situ sensors, and observed run-up height. Using satellite imagery, both[4,5] observe an initially 200 m run-up height near the location of the slide, with an average of 60 m run-ups being observed through the remainder of the fjord for the September 16th event.

Tsunami run-up for the October 11th even was only observed 200 m west of the gully ~75% of the magnitude of the September event in this location (60 m vs 80 m). Almost 72 km at the Ella $\varnothing$ research station, the initial run-up height was in excess of 4 km creating significant local damage. The location of Ella $\varnothing$ relative to the Dickson fjord is shown in Supplementary Fig. 1. To our knowledge, no information exists regarding the run-up at Ella $\varnothing$ for the October 11th event due to arctic winter darkness.

### Tidal modulation of the seiche

While the following results do not impact the conclusions made in this manuscript regarding the initial size of the seiche, they point to an interesting mechanism of seiche dissipation. The authors of ref. 4 observed an ~6 h modulation of the seiche frequency ranging from 10.874 to 10.879 mHz which they attribute to tidal influences (their Supplementary Fig. S1C). They suggest a possible mechanism is the change of depth modulating the speed of the wave $c$ through the relationship $c = \sqrt{gH}$ where $g$ is the gravitational acceleration and $H$ is the depth. However, a simple inspection of both the September and October VLP signals shows that the seiche actually behaves in the opposite manner (e.g., longer periods at higher tides). We here suggest an alternative mechanism; stratification. In semi-enclosed basins (defined as having an open-boundary for waves to radiate into the sea), the seiche wave speed is modulated by stratification through $\sqrt{gH\Delta\rho/\rho}$ where $\rho$ is the average density, and $\Delta\rho = \rho_2 - \rho_1$ is the density difference from top ($\rho_1$) to bottom ($\rho_2$)[36]. Inspection of the correlation between surface-water elevation and density $\rho$ shows a strong positive correlation (Supplementary Fig. 7). Based on this observation we proposed the following explanation. At low tide, the water column is strongly stratified with fresh (less dense) glacial runoff sitting on top, and denser salty water sitting beneath $\Delta\rho > 1$. As the tide comes in, the water column mixes, leading to the observed higher density $\rho$ at the surface and thus less stratification $\Delta\rho < 1$. Due to the fact that the fjord is quite deep $\approx 500$ m, the impact of $\Delta\rho \gg \Delta H$ as observed. Due to the fact that the CTD gauge only provides observations at the surface, it is impossible to conclude definitively that the density fluctuations observed are indicative of the stratification between $\rho_1$ and $\rho_2$. Hence, further simulated study is needed to validate this hypothesis.

## Data availability

All data used in the study have been deposited in a Zenodo repository alongside code to process and produce each figure https://doi.org/10.5281/zenodo.15166491[37]. The L2 HR SWOT data can be accessed freely through PO.DAAC 10.5067/SWOT-PIXC-2.0[38].

## Code availability

All code needed to replicate the given analysis can be found in the zenodo repository https://doi.org/10.5281/zenodo.15166491as well as the dedicated code ocean repository (https://codeocean.com/capsule/5272497/tree).

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

## Acknowledgements

T.M. and T.T. acknowledge support from the Eric and Wendy Schmidt AI in Science Postdoctoral Fellowship, a Schmidt Futures program.

## Author contributions

Conceptualization: T.M., Formal Analysis: T.M., Funding Acquisition: T.A., Investigation: T.M., Methodology: T.M., Supervision: T.T., S.R., T.A., Writing (original): T.M., Writing (review and editing): T.M., T.T., S.R., T.A.

## Competing interests

The authors declare no competing interests.
