## [Transparent Peer Review file · Nature Communications]

Observations of the seiche that shook the world

Corresponding Author: Mr Thomas Monahan

Version 0:

Reviewer comments:

Reviewer #1

(Remarks to the Author)

This manuscript attempts to resolve a seiche for the first from satellite observations using the recently launched wide-swath SWOT satellite data. The paper compares and correlates their findings with previous studies and attempts to resolve certain limitations within the SWOT data. Overall, the findings of this study are particularly interesting, and the results presented by the authors provide exciting insights into future works using this data to evaluate more seiches as they progress.

This manuscript is well written, especially considering the multiple disciplines that this article will be addressing. On this, a few more descriptions of terms from across the fields may help the readability of this manuscript, especially based on terms such as Rayleigh and Love, which have definitions in each of the fields discussed: altimetry, oceanography and seismology. The manuscript is publication-worthy, but I have a few recommendations that would potentially strengthen the arguments made by the author and increase the readability of this work.

Major point:

1. The usual suspects, Tides:

The authors attempt to account for tidal influence in the SWOT data to rule it out as a contributor to the signals being seen in the data. As a first step, the authors estimate the M2 tidal constituent and only use this to correct the data that they are analysing instead of relying on any model.

1a. The first problem with this is that they only justify the use of ~20 days worth of SWOT data, while the aliasing of SWOT in the Science orbit requires significantly more data than this (~66 days). The authors seem to generate OK results despite this compared to a CTD, and I suspect this is thanks to SWOT's orbit somewhat overruling aliasing periods OR the fact that the authors rely solely on the M2, which should be the easiest to capture. On inspection of the crosssections within Figure 4, the estimates may be 10s of centimetres off the nearby CTD and almost 20 degrees off the phase lag. In the explanations, they discuss only solving for M2 in the harmonic formula, which I think may be causing some of these larger errors. In early work on SWOT, we included several tidal constituents even when only interested in solving for M2 in an attempt to get a more 'pure' M2 signal that would be captured by the nearby observation. But due to the limited time period, I am not sure this is true. Did the authors test this?

1b. I don't think I agree with not using FES2022 as a correction for SWOT [Section 6.0.1. Data processing], also considering 1a. On analysis of the CTD station, M2 is definitely the biggest contribution (and I confirm the authors' calculations), but there are still tides observed that are big enough to be necessary, e.g. 23 cm, 10 cm, 8 cm and 7 cm for S2, N2, K1 and O1, respectively. The CTD hints at tidal ranges exceeding 1 meter comfortably, and M2 only accounts for about 40 cm; there is still quite a lot of tidal variability remaining. Although I suspect your analysis for the seiche won't be much different or your conclusions won't change drastically, it may influence the MXCS estimations made in section 2, but if you only resolve one tide, which is not the only driving factor of tidal variability, you are not 'ruling out the usual suspects' sufficiently. However, in the seiche analysis, I would suggest keeping the FES2022 OR further justifying why not to use it. In my comparison with the CTD, FES2022 is not that far off and also demonstrates amplitude changes from east to west within the fjord.

Minor points:

"This limited data is insufficient to resolve additional tides ..." Is there a reference to this, how much data is required? On this, if sufficient SWOT data was available, how would it refine your tidal estimation and then your seiche estimates? I.e. would this study improve in 5 years?

From section 3.1, it is also important to potentially reference that the M2 you are deriving is significantly seasonally driven (see Bij de Vaate et al 2021) because you are only able to obtain measurements in ice-free seasons which are likely only summer months.

The figure designs: The colourbar selection of Figure 4 needs to be changed, maybe to focus only on minimums and maximums within the data range (i.e. going up to 360 degrees is not fully necessary in this and maybe hides some small variations in the phase lag). Additionally, some of the figure labels and axis labels are difficult to read (except when you can zoom in a lot on the online versions).

When you analyse the time-series of gridded SWOT data that you create, does the mean fjord SLA demonstrate any signs of the seiche? Or is this not so easily seen in the SWOT data directly?

Figure A1, I think the CTD depth time-series should also be included to show its tidal influence.

What are the red lines in Figure A1 representing? SWOT overflights?

I would not limit this study to highlighting the importance of 'extremes'. Seiches are generally physical phenomena that we have struggled to observe from satellites, but they themselves are not only extreme events.

The motivation for future missions in the context of this study, although of course great, in the context of extremes, the need for higher temporal sampling outweighs a non-sun-synchronous orbit, for example. The sampling frequency of satellite altimetry is a huge limitation for extreme analysis.

For terms like Love and Rayleigh, they also have meaning in the tidal world. I understand this is potentially an obvious term for seismology, but a short context in this paper may help with readability outside seismology.

Do the authors account for the influence of sea ice within the SWOT data? What correction is used or is it manually done by the authors?

Section 6.0.4 and 6.0.5 are fine, maybe they require a bit more referencing.

Is the CTD mounted to the bottom of a mooring or on the surface?

Some of the references are incomplete (e.g. 31 and 32 have no journal).

Figure A2 and A3 needs a colour bar or a description of what the colours represent.

Bij de Vaate, I., Vasulkar, A.N., Slobbe, D.C. and Verlaan, M., 2021. The influence of Arctic landfast ice on seasonal modulation of the M2 tide. *Journal of Geophysical Research: Oceans*, 126(5), p.e2020JC016630.

MHD

(Remarks on code availability)

Reviewer #2

(Remarks to the Author)

This submitted manuscript offers a new study of the course of monochromatic globally observed seismic signals in Sept. and Oct. 2023. The authors combine SWOT data, seismic observations and Bayesian approaches to validate previous studies which attributed the monochromatic signal to a seiche produced by a landslide-generated tsunami in the Dickson Fjord, Greenland. The study presents the first direct observations of the seiche via SWOT satellite data. They estimate the initial amplitude of the wave to 7.9 m and demonstrate the importance of satellite altimetry data to study extreme geophysical events.

The manuscript starts with a comprehensive summary of the findings of two previous studies on the same events, which motivates nicely the usage of SWOT data for a more direct observation of the seiche. Then it presents the new dataset and its evidence for the proposed process, also confirming the estimates of wave heights from previous studies which were based on seismic waveform data at large distances. Through an analysis of wind speed and direction as well as one year-long tidal observations on SWOT data, other suspect causes of the water oscillation are ruled out. This is necessary, because while SWOT data provides more direct, new observations, it has the disadvantage of very low temporal sampling with only snapshots of the process.

The manuscript is in a good state with both clear figures and text that is easy to follow. The paper is contributing to the field of multi-disciplinary arctic research and it will be of interest to a broad audience. By summarizing and discussing previous findings as well as providing new evidence for the seiche, it is a nice round-up of studies of the very interesting and strange 2023 Dickson Fjord events. We recommend the acceptance of the manuscript after some minor revisions.

Comments:

- The description of the resolution/ uncertainties of the SWOT data is unclear. In Fig. 1C it says along-track resolution 2.5 m, cross-track resolution 70 m-10m. In 2.1 paragraph 2 it says 6 m along track and 10 m cross-track. In addition, it is the vertical resolution is not described at all, which is key for the water level estimates.

- In section 6.0.3 you describe how you selected the Rayleigh wave speed and which earth model you use, but have you checked the effects of different wave speeds and models?

- Carrillo-Ponce et al. observe a temporal modulation of the dominant frequency of the VLP signal (supplement of their paper). However, they note that the modulation of the frequency is opposed to what would be expected from a simple increase and decrease of the water level height inside the fjord system. Do you have any idea what could be causing this issue? (This is just a general question out of curiosity for the seiche process and the tides. It is likely beyond the scope of

their paper, but anyway an interesting point to think about.)

Minor comments / typos

- Just a comment: Line numbers would have been helpful for the reviewing process...
- Introduction, 1st paragraph: "scientist must increasingly depend on..." Why "increasingly"?
- Introduction, 1st paragraph: "...often necessary, resulting in..."  "potentially resulting in"?
- Check all references: For example Places like Greenland should always be capitalized (Greenland, not greenland).
- Section 3, 1st paragraph: "conform" or "confirm"?
- 6. Methods: Provide a Link for the SWOT data access
- 6.0.1 Data processing: Why are values that exceed 4m removed? Please clarify.
- 6.0.7 In Situ Measurements: Why is the seiche not observable on the tidal gauge? From it's location in the channel we would expect a signal to be seen especially in the early parts of the seiching when the wave height is highest.
- 6.0.8 "...climate of the can be found..."  "...climate of the region can be found..."?
- Fig. 3 caption: "H-K"  "G-J".

(Remarks on code availability)

We assessed the code, but it was not possible for us to run it on the ocean code server. We were however able to run the jupyter notebooks on figshare. On code ocean, we could see that the scripts have been run there by the authors, resulting in the figures and results shown in the manuscript.

The code is well commented and clearly structured, we think it is sufficient to reproduce the results. While a detailed analysis of bugs or flaws would require an in-depth usage of the code, the provided code will certainly help others to reproduce the processing also for other datasets.

Reviewer #3

(Remarks to the Author)

(Remarks on code availability)

Reviewer #4

(Remarks to the Author)

title: "Review of 'First Observations of the Seiche That Shook the World'"
date: "Jan 19, 2025"

General remarks

The study under review examines an unprecedented seismic phenomenon linked to extreme climatic events, specifically focusing on the seiche dynamics observed in the Dickson Fjord, Greenland, following two tsunamigenic landslides in September and October 2023.

Key Themes

1. **Emergence of Novel Phenomena**:

- The authors argue that climate change is driving the rise of new, previously unobserved events. In remote regions like the Arctic, the lack of in-situ measurements necessitates reliance on analytical and numerical modeling, though these methods face significant challenges due to data uncertainties.

2. **Focus on the Seiche Phenomenon**:

- Central to the discussion is the mysterious 10.88 mHz very-long-period (VLP) seismic signal, hypothesized to result from a standing wave (seiche) triggered by a landslide-induced tsunami. Notably, the study highlights the anomalous 9-day persistence of this signal, followed by its reemergence after a second landslide.

3. **Modeling and Observational Discrepancies**:

- Despite reproducing seismic amplitudes, analytical and numerical models yield differing estimates for the initial seiche amplitude, ranging from 2.6 m to 7.4–8.8 m. These discrepancies underscore the necessity for empirical validation to refine model assumptions and enhance understanding of such dynamics.

Evidence for the Seiche Hypothesis

- studies reviewed in the paper identify the Rayleigh and Love wave radiation pattern as consistent with a single oscillating force perpendicular to the fjord. Using different methodologies:
- **[4]**: Analytical modeling with simplified fjord geometry estimated an initial amplitude of 2.6 m.
- **[5]**: High-resolution numerical simulations estimated a larger initial amplitude of 7.4–8.8 m, corroborated by seismic waveform modeling.

While both studies convincingly argue for the seiche hypothesis, their differing assumptions regarding fjord geometry and other unmodeled effects contribute to the divergence in amplitude estimates.

Novel Contributions

This review appreciates the authors' efforts to integrate interdisciplinary evidence, including satellite imagery, seismic observations, and advanced numerical models, in addressing this complex phenomenon. The study's call for direct empirical observations to resolve model uncertainties is compelling, marking a toward a deeper understanding of seiche dynamics in remote and data-scarce regions.

Review

Full disclosure, I am an "applied" computational/statistical geophysicist who has not worked in this research area. This may in part may explain the difficulties I experienced working my way through this manuscript, which is rife in domain specific lingo whether this concerns the topic itself or the statistical analyses to back up the claims.

Since my background aligns with Bayesian analyses, I will limit my review to that aspect of the manuscript.

General remarks

Frankly, I find the theoretical analysis extremely difficult to follow for the following reasons

- mathematical symbols are not always defined properly neither are basic statements included on what is what---e.g. what is known what is unknown. It also unclear what are scalars, vectors, and matrices. What do the weights w represent? What is the y_i observation y_i ? What does the design matrix represent what do the sizes $M \times N$ refer to? Is Y a matrix or a vector. Same for the α . There is mention of α_k , which suggests it may be a vector but what does the subscript k refer to? What is β ? Text above equation 2 suggests it is a covariance matrix but unsure.
- The mathematical/statistical treatment is muddled and impossible to follow. Why is the design matrix included in the likelihood? Why is the factorization in equation 4 justified? What are the assumptions? What is meant with (natural) shrinkage? Why is the gamma distribution used to generate non-negative weights? What are mixing parameters? What are hyper-hyper parameters?
- Why is initialization with the ML (suggests a Laplace-type approach) justified? While the problem is linear, the complicated prior may make the problem non-convex---i.e., subject to local minima. There is no mention of this.
- Aside from computational benefits, what is the motivation to use VI?
- There are lots of mathematical inconsistencies and unexplained terminology. For instance, while equation 2 suggests that θ includes w and β equation 10 seems to suggest otherwise. Also, above equation 10 the authors refer to the rather technical term "conjugate with the prior" without explaining what it is and stating that this leads to a specific relationship between the prior and likelihood. Also, the sentence "This term provides natural model shrinkage as it increases with the number of free parameters θ ." is difficult to unpack. So is the ensuing text.
- Equation 13 is not an equation since it misses a left-hand side. Also, the subscript k is not defined (perhaps I missed it but then a repeat would be helpful) neither its range. It seems to me that the angular frequency (not Hz) ω misses this subscript when looking at equation 14. On p18 top ω is described in terms of deg/h, which strikes me as odd. I would have expected something with radians
- Unfortunately, I find the discussion surrounding equation 16 impossible to follow. I do not see how this akin a convolution, etc.

Overall, I have to say that while I absolutely appreciate efforts to make the computational aspects of this manuscript reproducible (by including code, etc.), the writeup of the technical part does from my perspective not meet the standards of this journal. It is muddled and leaves out important details and as a consequence can not be followed by the readership even if these are specialists in the area or in Bayesian statistics. For this reason, I recommend rejection of this manuscript.

Detailed remarks

- Figure 1, Panels B, D and E. In the text there is reference to these panels while this figure only includes B, C, D.
- the term "gradients" are nowhere defines. Gradient of what. Is this the same as "slope" also not defined. This makes it extremely difficult to follow what is going on.
- other "frequentest" variants p 14. I believe the authors may refer to "frequentist".
- "such as natural parameter shrinkage" is not defined. What does it mean?
- what does the subscript k on p 17 eq 13 refer to?
- missing k subscript for ω
- $\omega_{M2} = 28.985 \text{ deg/h}$ should'nt this be in radians?

- Consider the linear model given by $y_i = w^T x_i + \epsilon_i$. Define what is what. What does this linear model represent?
- β , which represents the precision, or inverse (co)variance of the noise. Not clear whether this is a scalar or a matrix
- Why condition the likelihood on $p(Y|X, \theta) = p(Y|X, w, \beta)$ X?
- The model weights $p(w|\alpha)$ come from a zero-mean Gaussian prior, with precision (inverse variance) α . What is the motivation for this?
- The set of weight precisions α , which govern the scale of the weights w , are drawn from a Gamma distribution, which models the distribution over non-negative precisions. Why this choice?
- Uninformative mixing hyper-parameters $a_0 = 10^{-2}$ and $b_0 = 10^{-4}$ are selected to yield a vague prior over each α_k defined as $p(\alpha_k) = \Gamma(\alpha_k; a_0, b_0)$. What is mixing? What is a vague prior? How does it compare to the uniform distribution on a positive interval?
- Models are initialized using a maximum-likelihood (ML) solution such that. Why is this a good initial guess for a problem that may have become non-convex due to the weights?
- "Fully Bayesian solutions are obtained by marginalizing over the posterior distributions of the parameters." What are you marginalizing over???
- The difficulty arises when computing the posterior distribution, which analytically is almost always intractable. Correct but is this also true for the special structure of your priors?
- The functional form of this posterior is chosen to be conjugate with the prior over θ such that $q(\theta|Y)$ factorizes as $q(\theta|Y) = q(w|Y)q(\alpha|Y)q(\beta|Y)$. This needs additional explanation.
- $P_{j,k}$ is given by the amplitude $w_{0,0}$ of the central point $P_{0,0}$ with a small offset denoted $w_{j,k}$. Too little detail.

(Remarks on code availability)

Version 1:

Reviewer comments:

Reviewer #1

(Remarks to the Author)

Dear Monahan et al

I appreciate the effort given by the authors to address my reviews as well as those of the other reviewers. I am satisfied with the responses and I feel the overall quality of the manuscript is much better than before. Particularly the additional work added regarding the tides is more valid than before and I am now happier with the conclusions that are being drawn. I think this manuscript is overall a good quality, highly relevant and suitable for publication.

With all that being said, I have nothing further to add. My only small comment, that I will leave to the authors, is for Figure A7, I would only show the validation of the tides that are common between both approaches.

Kind regards
Mike Hart-Davis

(Remarks on code availability)

I visually inspected the code, particularly around the tidal analysis. This is all legitimate. I didn't run the code directly.

Reviewer #2

(Remarks to the Author)

Dear editors, dear authors,

we have looked at all of the provided revised material and find that the authors have thoroughly responded to our questions. We have no further issues or questions.

As for the part of the potential tidal modulation of the oscillation frequency we think the authors can decide if they want to keep the section or not. They formulate one idea to explain the observation, but we feel it would need to be proven by some quantitative approach to confirm the hypothesis and it is nevertheless beyond the main research question of the paper.

(Remarks on code availability)

Reviewer #3

(Remarks to the Author)

(Remarks on code availability)

Reviewer #4

(Remarks to the Author)

Thanks for addressing the points I raised in my review. One minor point, in the text you write

> First, a Gaussian is the least informative distribution for a quantity which can be either positive or negative, and 526 is thus unbiased in this way [29].

In my understanding, the paper by Roberts et al. (2013) (ref [29]) does state that "The Gaussian is the least informative choice of distribution for a quantity that can be either positive or negative, and is therefore appropriate when...". However, this statement is context-dependent.

In information theory, the Gaussian (normal) distribution is considered the maximum entropy distribution for a given mean and variance, meaning it is the "least informative" distribution under those specific constraints. This implies that if you only know the mean and variance of a real-valued quantity, assuming a Gaussian distribution introduces the fewest additional assumptions. However, if no constraints on mean or variance are specified, the Gaussian distribution is not inherently the least informative choice. Therefore, the applicability of the statement from Roberts et al. (2013) depends on the specific context and constraints of the problem being addressed.

Having said that I recommend publication of this manuscript.

(Remarks on code availability)

Response to Reviewers: First observations of the seiche that shook the world

Thomas Monahan, Tianning Tang, Stephen Roberts, Thomas A. A. Adcock

We would like to thank all of the anonymous reviewers for their helpful feedback on our manuscript. By implementing these suggestions, we believe the revised manuscript is more comprehensive and understandable for interdisciplinary audiences. Detailed responses to each reviewer's comments are provided in the respective response to reviewer documents. A brief summary of the major changes is given below. We also note that all changes have been highlighted in the revised manuscript.

1. **Overhaul of the methods section:** We have added considerable detail and clarification to the Bayesian analysis and subsequent tidal analysis sections to make this more accessible to interdisciplinary readers. This includes reducing jargon, or explicitly defining terms when necessary.
2. **Improved tidal estimation and validation:** As suggested by Reviewer 1, the tidal analysis section has been significantly improved through (i) an improved analysis procedure, and (ii) access to more data. In addition to a revised Figure 4, we have added a tidal validation section (6.0.7) which validates our approach against both the in-situ CTD observations and the state-of-the-art assimilative tide model FES2022.
3. **Figure tweaks and added definitions:** All figures have been redone to improve legibility of labels. We have added definitions for phenomena which have different definitions between fields (e.g. Rayleigh and Love waves).
4. **Additional note in supplementary materials on tidal modulation of the seiche** As reviewer 2 pointed out, and is discussed in [1], there is a modulation of the seiche period which appears to be related to tides (e.g. 6 hour period), but contrary to our expectation of a wave speed of \sqrt{gH} . Through analysis of the cross-correlation between surface-water density and elevation we argue this discrepancy is likely due to changing stratification. We would like to stress that we were unsure whether to include this in this manuscript, as it is not needed to justify our conclusions regarding the initial seiche behavior. However, we do believe it points to an interesting aspect of seiche dissipation. If the reviewers feel this is superfluous for this manuscript, we can omit.

References

- [1] Carrillo-Ponce, A. *et al.* The 16 September 2023 Greenland Megatsunami: Analysis and Modeling of the Source and a Week-Long, Monochromatic Seismic Signal. *The Seismic Record* **4**, 172–183 (2024).

Response to Reviewer 1: First observations of the seiche that shook the world

1 Overview

We would like to thank the reviewer for their extensive comments, and the helpful suggestions regarding the tidal aspects of our manuscript. We have implemented all of the proposed changes which has greatly strengthened the strength of our tidal analysis section. The reviewer's comments are given below in bold, our responses immediately follow.

1.1 Major Point

1. The usual suspects, Tides: The authors attempt to account for tidal influence in the SWOT data to rule it out as a contributor to the signals being seen in the data. As a first step, the authors estimate the M2 tidal constituent and only use this to correct the data that they are analysing instead of relying on any model. 1a. The first problem with this is that they only justify the use of 20 days worth of SWOT data, while the aliasing of SWOT in the Science orbit requires significantly more data than this (66 days). The authors seem to generate OK results despite this compared to a CTD, and I suspect this is thanks to SWOTs orbit somewhat overruling aliasing periods OR the fact that the authors rely solely on the M2, which should be the easiest to capture. On inspection of the crosssections within Figure 4, the estimates may be 10s of centimetres off the nearby CTD and almost 20 degrees off the phase lag. In the explanations, they discuss only solving for M2 in the harmonic formula, which i think may be causing some of these larger errors. In early work on SWOT, we included several tidal constituents even when only interested in solving for M2 in an attempt to get a more 'pure' M2 signal that would be captured by the nearby observation. But due to the limited time period, I am not sure this is true. Did the authors test this? 1b. I don't think I agree with not using FES2022 as a correction for SWOT [Section 6.0.1. Data processing], also considering 1a. On analysis of the CTD station, M2 is definitely the biggest contribution (and I confirm the authors' calculations), but there are still tides observed that are big enough to be necessary, e.g. 23 cm, 10 cm, 8 cm and 7 cm for

S2, N2, K1 and O1, respectively. The CTD hints at tidal ranges exceeding 1 meter comfortably, and M2 only accounts for about 40 cm; there is still quite a lot of tidal variability remaining. Although I suspect your analysis for the seiche won't be much different or your conclusions won't change drastically, it may influence the MXCS estimations made in section 2, but if you only resolve one tide, which is not the only driving factor of tidal variability, you are not 'ruling out the usual suspects' sufficiently. However, in the seiche analysis, I would suggest keeping the FES2022 OR further justifying why not to use it. In my comparison with the CTD, FES2022 is not that far off and also demonstrates amplitude changes from east to west within the fjord.

We are grateful to the reviewer for this extensive feedback and the suggestions for improving the tidal analysis in our manuscript. We have now redone the tidal analysis section (and added a new validation section) to take all of these suggestions into account. First, we now include multiple additional constituents (N2,S2,K1,O1) in the analysis to achieve a more accurate estimate of M2. We settled on this combination of constituents after considerable testing. The resultant M2, is much more accurate and yields significant improvements in RMS error over FES2022 as shown in the new Tidal Validation section. Second, in addition to our SWOT computed M2, we now utilize all of the FES2022 constituents which have less than 20% relative RMS error (see Tidal Validation Section). As before the residual tidal variability is accounted for in the MXCS computation by adaptively setting the midpoint. Using the revised M2, we can say with much higher confidence that there is no observable cross-channel variability. A more comprehensive description of these revisions is given for each of the minor points below. We also refer the reviewer directly to the revised (and highlighted) Sections 6.0.1, 6.0.6,6.0.7.

1.2 Minor points:

“This limited data is insufficient to resolve additional tides...” Is there a reference to this, how much data is required? On this, if sufficient SWOT data was available, how would it refine your tidal estimation and then your seiche estimates? I.e. would this study improve in 5 years?

We have now added a new section which directly compares the derived SWOT and FES2022 tides with those of the in-situ CTD Gauge. The new section, entitled “Tidal Validation”, provides a nice justification for (i) using the M2 tide from SWOT, and (ii) reasons for including some, but not all of the FES tides. Within this section we also discuss the future implications of future improvements in tidal estimation for this study as follows: “As more SWOT data becomes available, the accuracy of tidal estimates will improve. It is unlikely that these improved estimates will change the conclusions made by this study as we are primarily interested in the cross-channel slope, and find no evidence of this being tidal driven from inspection of the M2 tide. However, improved tidal estimation could prove useful for understanding how the tide itself modulates the period of the seiche, and thus impacts its dynamics and dissipation.”

From section 3.1, it is also important to potentially reference that the M2 you are deriving is significantly seasonally driven (see Bij de Vaate et al 2021) because you are only able to obtain measurements in ice-free seasons which are likely only summer months.

This is a great point and we have added the following section to discuss this: “It is important to note that there are large seasonal changes in tides within arctic regions [1]. Due to the limited SWOT data available, and the relatively short CTD gauge deployment (≈ 2 months), the extent of this variability is difficult to estimate. However, since the CTD gauge was deployed during both events, validation with this gauge is appropriate as we wish to have accurate tidal estimates during this period.”

The figure designs: The colourbar selection of Figure 4 needs to be changed, maybe to focus only on minimums and maximums within the data range (i.e. going up to 360 degrees is not fully necessary in this and maybe hides some small variations in the phase lag). Additionally, some of the figure labels and axis labels are difficult to read (except when you can zoom in a lot on the online versions).

We appreciate these suggestions and have implemented them accordingly. The tidal figure (Figure 4) now utilizes new, and much tighter colormaps which allow viewers to clearly see variations in phase throughout the fjord. Regarding the other figures, we have attempted to increase the fontsize for figure/axis labels where possible for Figures 1-3. We hope that these modifications will improve the legibility of our figures.

When you analyse the time-series of gridded SWOT data that you create, does the mean fjord SLA demonstrate any signs of the seiche? Or is this not so easily seen in the SWOT data directly? This is quite an interesting question. After some testing, no, the mean SLA of the entire fjord does not exhibit signs of the seiche. We believe this is a consequence of the sloshing motion of the seiche averaging out when one considers the mean. In contrast, if one considers the variance of the SLA, the seiche is readily seen (e.g. higher variances). However, both procedures are sensitive to the noise within the data and retracking errors that may be present which makes autonomous applications difficult. We have added the following note in the conclusion to stress this as an area of future work: “Additionally, we find that even when accurate SLA estimates can be obtained, the mean SLA of the fjord does not exhibit signs of seiching. This underscores the importance of accounting for even small-scale spatial variances, which SWOT and future wide-swath altimetry missions have the potential to address.”

Figure A1, I think the CTD depth time-series should also be included to show its tidal influence.

We have now added an additional Figure, (Figure A6 in the revised manuscript) which provides a complete view of the CTD depth observations alongside the non-tidal residuals during the two events. This figure has been included to extend our discussion on the tidal contributions in the Dickson Fjord, and to further justify the fact that the seiche quickly recedes beneath the CTD noise floor after a few hours.

What are the red lines in Figure A1 representing? SWOT overflights?

We appreciate the reviewer pointing out this as we had not labeled these in the original figure. We have now added a description to the figure which reads “The red dashed vertical lines indicate when SWOT observations occur.”

I would not limit this study to highlighting the importance of ‘extremes’. Seiches are generally physical phenomena that we have struggled to observe from satellites, but they themselves are not only extreme events.

We agree, the conclusion has been modified as follows “This study highlights the value of wide-swath satellite altimetry in characterizing extreme events and oceanic phenomena more generally. Due to their short-periods, seiches have long been difficult to study from conventional altimeters. The spatial resolution of SWOT provides new opportunities in this area, as well as for studying other fast-moving oceanic processes such as storm surge, and even large waves. As shown, these data also provide the opportunity to connect and understand the complex interactions between climate change and the different components of the geosphere. However, this work”

The motivation for future missions in the context of this study, although of course great, in the context of extremes, the need for higher temporal sampling outweighs a non-sun-synchronous orbit, for example. The sampling frequency of satellite altimetry is a huge limitation for extreme analysis.

This is a great point and we have added the following point immediately after the previous point “Above all, designing orbits to achieve higher temporal sampling at lower latitudes should be prioritized as it presently limits the study of extremes.”

For terms like Love and Rayleigh, they also have meaning in the tidal world. I understand this is potentially an obvious term for seismology, but a short context in this paper may help with readability outside seismology.

We appreciate the reviewer pointing this out as it has the potential to be very confusing. We have now added a footnote after the first mention of Love/Rayleigh waves which formally defines these and the type of seismic motion they produce.

Do the authors account for the influence of sea ice within the SWOT data? What correction is used or is it manually done by the authors?

Sea-ice is handled through manual inspection of the SWOT data. We found this necessary as many standard metrics e.g. backscatter, or ice-flags were not functioning properly in the Fjord. We believe this is a retracking issue with the default processing.

Section 6.0.4 and 6.0.5 are fine, maybe they require a bit more referencing.

We are glad the original sections 6.0.4, 6.0.5 were satisfactory. We note that we have added considerable detail to this section on the request of reviewer 3, and believe the new section is more easily understandable to non-experts.

Is the CTD mounted to the bottom of a mooring or on the surface?

After corresponding with the owners/maintainers of the CTD gauge, we have added

the following sentence: “The CTD is mounted on the fjord wall, inside a HDPE tube to protect it from ice in winter.”

Some of the references are incomplete (.e. 31 and 32 have no journal). We have gone through and fixed all references, including protecting capitals and adding journals to those missing.

Figure A2 and A3 needs a colour bar or a description of what the colours represent. Figures A2, and A3 now both have colorbars to illustrate the variance of observed sea-surface heights.

References

- [1] Bij de Vaate, I., Vasulkar, A., Slobbe, D. & Verlaan, M. The influence of Arctic landfast ice on seasonal modulation of the M2 tide. *Journal of Geophysical Research: Oceans* **126**, e2020JC016630 (2021).

Response to Reviewer 2: First observations of the seiche that shook the world

1 Overview

We would like to thank the reviewer for their helpful comments and interesting question regarding the tidal modulation of the seiche. All suggestions have been implemented and are described below. The reviewer's comments are given in bold, our responses immediately follow.

2 Comments:

- The description of the resolution/ uncertainties of the SWOT data is unclear. In Fig. 1C it says along-track resolution 2.5 m, cross-track resolution 70 m-10m. In 2.1 paragraph 2 it says 6 m along track and 10 m cross-track. In addition, it is the vertical resolution is not described at all, which is key for the water level estimates.

We appreciate the reviewer pointing this out as our original presentation was unclear. With regards to the vertical resolution, we have added the following note "These measurements have extremely low instrumental noise of less than 0.4 cm [1]." For the horizontal resolution, the correct specifications have been updated in the text as follows: "SWOT provides high accuracy measurements directly up-to coastlines, and uniquely into Fjords, with an effective pixelcloud resolution of 2.5 m along-track and a variable resolution ranging from 10 m to 70 m in the cross-track direction [2]. These measurements have extremely low instrumental noise of less than 0.4 cm [1]." It is worth noting that the noise seen in the SWOT measurement likely comes from retracking issues with the SWOT processing algorithms and not the measurement device itself. This is the reason our specialized procedure for data reprocessing is needed.

- In section 6.0.3 you describe how you selected the Rayleigh wave speed and which earth model you use, but have you checked the effects of different wave speeds and models?

The choice of Rayleigh wave speed has significant implications for the alignment of the SWOT observations with the seismic data. This serves as a primary motivation for (i) using the Earth model which was found to be most accurate by [3], and (ii) independently and empirically validating the derived wave speed. As was found in

both [3, 4], other models yielded poor estimates of the VLP source location. We found these models yielded similarly poor results for the velocity estimation and were thus not included. We added the following sentence to reflect this: “Additionally, testing of homogeneous Earth models yielded different estimates of both source location and velocities, yielding variable results. We found the agreement between these models and the empirically estimated wave travel time to generally be worse and thus did not consider them further.”

- Carrillo-Ponce et al. observe a temporal modulation of the dominant frequency of the VLP signal (supplement of their paper). However, they note that the modulation of the frequency is opposed to what would be expected from a simple increase and decrease of the water level height inside the fjord system. Do you have any idea what could be causing this issue? (This is just a general question out of curiosity for the seiche process and the tides. It is likely beyond the scope of their paper, but anyway an interesting point to think about.)

This is a very interesting point, and one which we did some preliminary analysis of. As noted by Carrillo-Ponce et al., the modulation appears to have a 6 hour period, but it seems unlikely this would significantly change \sqrt{gH} and hence change the speed of the seiche wave. Due to the depth of the fjord, we do not find this surprising. We instead propose an alternative mechanism; stratification. We describe this theory, and preliminary validation in the new Section 6.0.11 ‘Tidal Modulation of the Seiche’. We would like to stress that while this is an interesting result, we are unsure whether it is relevant to include within this manuscript. If the reviewer feels it is best to leave out then this can be omitted and written up separately.

2.1 Minor comments / typos

- Just a comment: Line numbers would have been helpful for the reviewing process...

We apologize numbers were not included in the first draft, these are now included in the revisions.

- Introduction, 1st paragraph: ”scientist must increasingly depend on...” Why ”increasingly”?

We have revised this to instead say “...scientists depend on...”. Thank you for this suggestion.

- Introduction, 1st paragraph: ”...often necessary, resulting in...” -¿ ”potentially resulting in”?

We have amended this sentence as suggested.

- Check all references: For example Places like Greenland should always be capitalized (Greenland, not greenland).

Thanks for pointing this out, all references have now been fixed/capitals protected.

- Section 3, 1st paragraph: "conform" or "confirm"?

Conform is intentional here as the observed cross-channel slopes agree with our expectations.

- 6. Methods: Provide a Link for the SWOT data access

We now provide a direct link to the SWOT Pixelcloud data within the data and processing methods section.

- 6.0.1 Data processing: Why are values that exceed 4m removed? Please clarify.

We have added the following explanation for removing values in excess of 4 m.

“This threshold was empirically determined following manual inspection of the dataset employed for tidal estimation. Due to retracking issues, spurious observations exist over the Fjord which exceed this 4 m threshold and are thus filtered out to avoid biasing estimates.”

- 6.0.7 In Situ Measurements: Why is the seiche not observable on the tidal gauge? From it's location in the channel we would expect a signal to be seen especially in the early parts of the seiching when the wave height is highest.

The CTD gauge has a sampling interval of 15 minutes which strongly aliases the seiche signal (period approximately 88 seconds). We have added an additional Figure A6, which shows the CTD gauge observations during both events alongside the non-tidal residual. While the seiche is clearly observable in these signals, the seiche signal quickly receded beneath pre-event noise levels after a couple of hours. Due to the combination of (i) slight variations in the signal period, and (ii) noise from other non-tidal and non-seiche processes, we could not elicit meaningful information from the seiche from these data over this short period.

- 6.0.8 "...climate of the can be found..." –¿ "...climate of the region can be found..."?

This has been fixed to say “climate of the fjord system”. Thank you for catching this.

- Fig. 3 caption: "H-K" –¿ "G-J".

We appreciate the reviewer catching this, we have fixed it in the revised text.

References

- [1] Fu, L.-L. *et al.* The surface water and ocean topography mission: A breakthrough in radar remote sensing of the ocean and land surface water. *Geophysical Research Letters* **51**, e2023GL107652 (2024).
- [2] Morrow, R. *et al.* Global observations of fine-scale ocean surface topography with the surface water and ocean topography (SWOT) mission. *Frontiers in Marine Science* **6**, 232 (2019).

- [3] Svennevig, K. *et al.* A rockslide-generated tsunami in a Greenland fjord rang Earth for 9 days. *Science* **385**, 1196–1205 (2024).
- [4] Carrillo-Ponce, A. *et al.* The 16 September 2023 Greenland Megatsunami: Analysis and Modeling of the Source and a Week-Long, Monochromatic Seismic Signal. *The Seismic Record* **4**, 172–183 (2024).

Response to Reviewer 3: First observations of the seiche that shook the world

1 Overview

We would like to thank reviewer 3 for their time and effort in co-reviewing our manuscript.

Response to Reviewer 4: First observations of the seiche that shook the world

1 Overview

We would like to thank the reviewer(s) for their extensive feedback and questions regarding the methods of our study. We appreciate that the original methods section was difficult to follow and have made considerable efforts to rectify this. We have implemented all of the provided feedback and feel that the revised methods section is both more comprehensive and understandable to a general audience. Due to the methods employed for this study encompassing multiple disciplines, we have made every attempt to keep the presentation accessible. However, for some concepts, a complete description of the basic theory is beyond the scope of this write-up. We have made appropriate references to foundational texts in these instances. Reviewer comments are provided below in bold with our responses following immediately. All revisions are highlighted in the text.

2 Response to General Remarks

Mathematical symbols are not always defined properly neither are basic statements included on what is what—e.g. what is known what is unknown. It also unclear what are scalars, vectors, and matrices. What do the weights w represent? What is the i^{th} observation y_i ? What does the design matrix represent what do the sizes $M \times N$ refer to? Is Y a matrix or a vector. Same for the α . There is mention of α_k , which suggests it may be a vector but what does the subscript k refer to? What is β ? Text above equation 2 suggests it is a covariance matrix but unsure.

We appreciate this feedback as our original presentation did not make these items clear. In the revised Bayesian analysis section, we have gone through to ensure that all terms are clearly defined upon their first introduction and that their form (scalar, vector, matrix) is explicitly given. Most of these points are also addressed individually below.

- The mathematical/statistical treatment is muddled and impossible to follow. Why is the design matrix included in the likelihood? Why is the factorization in equation 4 justified? What are the assumptions? What is meant with (natural) shrinkage? Why is the gamma distribution used to

generate non-negative weights? What are mixing parameters? What are hyper-hyper parameters?

We appreciate the reviewer pointing out the difficulties in understanding these terms in the first version of the manuscript. All of these terms have now been defined in the new iteration, as well as justifications for the various distributions employed for analysis (e.g. Gaussian, Gamma, etc.). All of these points are addressed individually in the detailed reviews section and highlighted within the revised methods section.

- Why is initialization with the ML (suggests a Laplace-type approach) justified? While the problem is linear, the complicated prior may make the problem non-convex—i.e., subject to local minima. There is no mention of this.

The ML solution provides an initial estimate that follows from the likelihood, ensuring that we start in an informative region of parameter space. This is useful for initializing the noise precision, β , and helps the variational updates from settling into poor local minima. While absolute guarantees depend on problem-specific properties, ML initialization is a well-established heuristic that improves the stability and efficiency of variational inference. You are quite right about non-convexity. The problem arises in even linear models with regularizers which are non-isotropic, i.e. they avoid assuming that the imprecision associated with each parameter in the model is the same. This creates a problem that can no longer be solved using convex optimization and an iterative inference approach (such as VB) is required.

- Aside from computational benefits, what is the motivation to use VI?

Variational Bayes allows us to perform (albeit approximate) fully Bayesian inference. This means that not only does it entertain full probability distributions over the parameters of the model, it also performs Bayesian inference over the (hyper-) parameters that govern those distributions. This means that uncertainty associated with all parameters and hyper-parameters in the model is taken into account. This is a crucial benefit of full Bayesian inference. Alternative methods rely on sampling (such as MCMC), which becomes computationally prohibitive in anything more than small problems as well as having unreliable convergence. VB is thus a computationally tractable alternative to MCMC, bringing both scalability and a principled approach to assess whether convergence has been achieved. These attributes are particularly important for the tidal estimation procedure, as with over 300,000 individual locations, MCMC and the associated overhead of assessing convergence manually, is intractable.

We have added the following description immediately after the introduction of VBayes: “VB is a computationally tractable alternative to MCMC, bringing both scalability and a principled approach to assess whether convergence has been achieved. These attributes are particularly important for the tidal estimation procedure, as with over 300,000 individual locations, assessing convergence manually is intractable.”

- There are lots of mathematical inconsistencies and unexplained terminology. For instance, while equation 2 suggests that θ includes w and

β equation 10 seems to suggest otherwise. Also, above equation 10 the authors refer to the rather technical term "conjugate with the prior" without explaining what it is and stating that this leads to a specific relationship between the prior and likelihood. Also, the sentence "This term provides natural model shrinkage as it increases with the number of free parameters θ ." is difficult to unpack. So is the ensuing text.

We have revised the full Bayesian analysis section with these components in mind. Specifically, we have tried to added definitions and other detail to ensure the materials are effectively conveyed to experts outside of Bayesian analysis. All of the points mentioned here are individually addressed in the detailed comments below. However, we would like to note that the likelihood (Equation 2) and the approximate posterior q (Equation 10) are not at odds with one another. The likelihood reflects the likelihood of the observed data based on our model. In this case, the only relevant parameters are w and β . While the weight precisions α impact the weights through the inference step, they are not included in the likelihood. The posterior in contrast is composed of both the prior, and likelihood which are related through Bayes' theorem. Computation of the posterior must therefore include the posteriors of all individual elements of $\theta = [w, \alpha, \beta]$.

- Equation 13 is not an equation since it misses a left-hand side. Also, the subscript k is not defined (perhaps I missed it but then a repeat would be helfull) neither its range. It seems to me that the angular frequency (not Hz) ω misses this subscript when looking at equation 14. On p18 top ω is described in terms of deg/h, which strikes me as odd. I would have expected something with radians

We have fixed equation 13, to relate the quadrature amplitudes A_k, B_k to the constituent amplitude C_k and phase ϕ_k . k is also explicitly defined including the range (up to n constituents). The point about the phase speed ω is addressed in the detailed remarks and has been fixed also.

- Unfortunately, I find the discussion surrounding equation 16 impossible to follow. I do not see how this akin a convolution, etc.

We have now completely redone the description of the spatially coherent inductive bias and modified the text regarding the convolution. Since this procedure was originally proposed and is discussed in [1], our original presentation was written up quite sparsely. However, now all information required to replicate the approach and a more comprehensive discussion is provided. We hope this is more clear.

2.1 Response to Detailed Remarks

- Figure 1, Panels B, D and E. In the text there is reference to these panels while this figure only includes B, C, D.

We have gone through to ensure that there is no misreferencing of panels within the text for all of the Figures.

- the term "gradients" are nowhere defines. Gradient of what. Is this the

same as "slope" also not defined. This makes it extremely difficult to follow what is going on.

The term gradient was used to describe the visual characteristics of the sea-surface height maps which are being compared. We appreciate the reviewer pointing out that this term was not clear. We have revised the discussion to instead say "To obtain perceptually uniform sea-surface height maps which can be compared between passes, we set the center of each colormap to be the midpoint of the cross-channel slope." We hope this new description is more clear.

- other "frequentest" variants p 14. I believe the authors may refer to "frequentist".

This has now been fixed, we appreciate the reviewer pointing this out.

- "such as natural parameter shrinkage" is not defined. What does it mean?

We have now defined shrinkage explicitly where it is first introduced. The description reads as "Shrinkage refers to reducing the magnitude of parameters which do not contribute to the solution. This favors simpler models unless the data justify additional complexity."

- what does the subscript k on p 17 eq 13 refer to?

k simply refers to the k^{th} constituent, however, this was slightly ambiguous in our original presentation. We have revised this to say: "For the k^{th} constituent with frequency ω_k , the corresponding tidal wave is given in quadrature by..."

- missing k subscript for ω

This has been fixed, thank you for pointing it out.

- $\omega_{M2} = 28.985$ deg/h should'nt this be in radians?

ω_{M2} was originally given in deg/h as this is a standard way of expressing the "phase speed" of tidal constituents. However, since we have now redone the tidal estimation section, we have revised this to instead refer to the period of the constituents in hours. Since we now include several constituents in the analysis, these are given in the new Table A2.

- Consider the linear model given by $y_i = w^T x_i + \epsilon_i$. Define what is what. What does this linear model represent?

The linear model looks to model the SWOT observed sea-surface heights Y based on the functional form given by the design matrix X . We have considerably added to the description of this basic model. The revised description makes explicit what each element of the linear model is, and what the elements of the weight vector w are. These components are then described at length in the subsequent section of Bayesian Analysis and tidal analysis.

- β , which represents the precision, or inverse (co)variance of the noise. Not clear whether this is a scalar or a matrix

We appreciate the reviewer pointing this out, as the dimensionality of β was not clearly defined. We have modified the initial discussion of beta to say: “Here, β is a scalar value. Discussion and validation of this modeling decision in the context of SWOT based tidal analysis is provided in [1].” For context, we found it necessary to treat β as a scalar value due to the temporal sparsity of the SWOT data. In previous work, [1] found non-Gaussian residuals (e.g. those employed in iteratively-reweighted least squares) to significantly reduce model performance for these data. If the reviewer thinks this would be a useful discussion to add, we can certainly do so in the next iteration.

- Why condition the likelihood on $p(Y|X, \theta) = p(Y|X, w, \beta)X$?

We are unsure which aspect of the likelihood this question is asking about so what follows is a general overview of the likelihood chosen. In the context of variational Bayes, we condition the likelihood on both X (the design matrix) and the parameters w and β , as the model aims to estimate the posterior distribution of the parameters given the data. It is necessary to include the design matrix, as this fundamentally relates the parameters w, β to the data Y . In this way, the likelihood $p(Y|X, w, \beta)$ describes how the observed data Y is generated from both the inputs X and the model parameters w, β , and the objective is to infer the posterior distribution of these parameters via Bayes theorem.

- The model weights $p(w|\alpha)$ come from a zero-mean Gaussian prior, with precision (inverse variance) α . What is the motivation for this?

We believe our initial technical justification of the zero-mean Gaussian prior sufficiently justifies this decision. We have added an additional sentence which clarifies that having a prior which is unbiased for positive/negative values is necessary for both tidal and cross-channel slope estimation.

- The set of weight precisions α , which govern the scale of the weights w , are drawn from a Gamma distribution, which models the distribution over non-negative precisions. Why this choice?

We have added the following explanation “The Gamma hyperprior is conjugate with the Gaussian prior over the weights which will be useful when performing inference. A Gamma prior also implicitly encourages smaller weight magnitudes, leading to natural shrinkage and promoting sparsity in the inferred weights.” We have also added a footnote which explicitly defines what ‘conjugate’ means which reads: “A conjugate prior for a given likelihood function is a prior that, results in a posterior distribution that is of the same family as the prior.”

- Uninformative mixing hyper-hyper-parameters $a_0 = 102$ and $b_0 = 104$ are selected to yield a vague prior over each α_k defined as $p(k) = \Gamma(\alpha_k; a_0, b_0)$. What is mixing? What is a vague prior? How does it compare to the uniform distribution on a positive interval?

We have now added a note which clarifies that “...mixing hyper-hyper-parameters determine the shape and scale of the associated distribution.” As we defined in the

original text, “A *vague* or *uninformative* prior simply means the assumed distribution is broad.” We have now added several sentences which clarify why this is favorable over a uniform distribution.

- Models are initialized using a maximum-likelihood (ML) solution such that. Why is this a good initial guess for a problem that may have become non-convex due to the weights?

The ML solution provides an initial estimate that follows from the likelihood, ensuring that we start in an informative region of parameter space. This is useful for initializing the noise precision, β , and prevents variational updates from diverging or settling into poor local minima. While absolute guarantees depend on problem-specific properties, ML initialization is a well-established heuristic that improves the stability and efficiency of variational inference, particularly in cases where latent weights introduce non-convexity.

- ”Fully Bayesian solutions are obtained by marginalizing over the posterior distributions of the parameters.” What are you marginalizing over???

The marginalization is performed over the posterior distributions of the parameters as said. Since we use conjugate priors, the posterior maintains the same form, allowing for analytical marginalization. In variational Bayes, the iterative updates approximate this marginalization within the chosen analytical framework. We chose to present the information in this manner as it first outlines what the problem is, why classical approaches such as MCMC are insufficient, and then naturally flows into how variational inference overcomes this. If the revised presentation is still confusing, then we can reverse the order of this in the next iteration.

- The difficulty arises when computing the posterior distribution, which analytically is almost always intractable. Correct but is this also true for the special structure of your priors?

No, this is the exact reason for using conjugate priors! We have added an extended discussion on the mean-field approximation (the reason we assume the parameter posteriors can be factorized) and how this leads to analytical tractability.

- The functional form of this posterior is chosen to be conjugate with the prior over θ such that $q(\theta|Y)$ factorizes as $q(\theta|Y) = q(w|Y)q(\alpha|Y)q(\beta|Y)$. This needs additional explanation.

We have added the following note which discusses both why this is a reasonable choice, and the potential limitations it introduces: “This choice follows from the mean-field approximation, which assumes our latent parameters to be mutually independent. This selection is deliberate, as it allows for analytical marginalization over the parameter posteriors, making inference computationally efficient. However, it also neglects correlations between parameters, potentially underestimating uncertainty in the presence of strong multicollinearity. Despite this, the approach has been extensively validated for both trend extraction and tidal estimation and is thus sufficient for these tasks [1, 2].”

- $P_{j,k}$ is given by the amplitude $w_{0,0}$ of the central point $P_{0,0}$ with a small offset denoted $w_{j,k}$. **Too little detail.**

We appreciate the reviewer pointing this out. Our original presentation wrote this up concisely as its full development and validation was given in [1]. However, we agree that too few details were given to replicate it from the text (despite the inclusion of our code). We have completely redone this section to discuss the full implementation.

References

- [1] Monahan, T., Tang, T., Roberts, S. & Adcock, T. A. Tidal and mean sea surface corrections from and for SWOT using a spatially coherent variational Bayesian harmonic analysis. *Authorea Preprints* (2024).
- [2] Roberts, S., McQuillan, A., Reece, S. & Aigrain, S. Astrophysically robust systematics removal using variational inference: application to the first month of Kepler data. *Monthly Notices of the Royal Astronomical Society* **435**, 3639–3653 (2013).

Response to reviewers: Observations of the seiche that shook the world

Thomas Monahan, Tianning Tang, Stephen Roberts, Thomas A. A. Adcock

We would like to thank all of the reviewers for their time and for recommending acceptance of our publication. The following are responses to the three remaining pieces of feedback.

Reviewer 1:

"With all that being said, I have nothing further to add. My only small comment, that I will leave to the authors, is for Figure A7, I would only show the validation of the tides that are common between both approaches."

We considered including only the M2, S2, N2, K1, and O1 tides in Figure A7, however, we felt it to be useful to include the additional comparisons of just FES2022 to justify not including these other constituents in the analysis. Omission of this comparison would leave our claim in the methods unsubstantiated.

Reviewer 2:

"As for the part of the potential tidal modulation of the oscillation frequency we think the authors can decide if they want to keep the section or not. They formulate one idea to explain the observation, but we feel it would need to be proven by some quantitative approach to confirm the hypothesis and it is nevertheless beyond the main research question of the paper."

There is considerable interest in the broader community in modeling the seiche and we feel that our hypothesis offers an important contribution to the discourse in this area, namely in motivating the need to run models using stratification. As a compromise, we have moved this figure to the supplementary information and highlighted that the claim requires further study.

Reviewer 3:

"In my understanding, the paper by Roberts et al. (2013) (ref [29]) does state that "The Gaussian is the least informative choice of distribution for a quantity that can be either positive or negative, and is therefore appropriate when...". However, this statement is context-dependent.

In information theory, the Gaussian (normal) distribution is considered the maximum entropy distribution for a given mean and variance, meaning it is the "least informative" distribution under those specific constraints. This implies that if you only know the mean and variance of a real-valued quantity, assuming a Gaussian distribution introduces the fewest additional assumptions. However, if no constraints on mean or variance are specified, the Gaussian distribution is not inherently the least informative choice. Therefore, the applicability of the

statement from Roberts et al. (2013) depends on the specific context and constraints of the problem being addressed.”

We appreciate the reviewer’s clarification regarding the context-dependent nature of the maximum entropy property of the Gaussian distribution. We fully agree that the statement from Roberts et al. (2013) applies specifically under the assumption of known mean and variance constraints – as is standard in information-theoretic arguments.

In our case, this was precisely the context in which we were invoking the result: to justify the Gaussian assumption when modeling tidal quadrature amplitudes and cross-channel slopes which lack further information beyond the first two moments.